# Inverse Reinforcement Learning from a Gradient-based Learner

**Giorgia Ramponi**
Politecnico Di Milano
Milan, Italy
giorgia.ramponi@polimi.it

**Gianluca Drappo**
Politecnico Di Milano
Milan, Italy
gianluca.drappo@mail.polimi.it

**Marcello Restelli**
Politecnico Di Milano
Milan, Italy
marcello.restelli@polimi.it

## Abstract

Inverse Reinforcement Learning addresses the problem of inferring an expert's reward function from demonstrations. However, in many applications, we not only have access to the expert's near-optimal behaviour, but we also observe part of her learning process. In this paper, we propose a new algorithm for this setting, in which the goal is to recover the reward function being optimized by an agent, given a sequence of policies produced during learning. Our approach is based on the assumption that the observed agent is updating her policy parameters along the gradient direction. Then we extend our method to deal with the more realistic scenario where we only have access to a dataset of learning trajectories. For both settings, we provide theoretical insights into our algorithms' performance. Finally, we evaluate the approach in a simulated GridWorld environment and on the MuJoCo environments, comparing it with the state-of-the-art baseline.

## 1  Introduction

Inverse Reinforcement Learning (IRL) [20] aims to infer an expert's reward function from her demonstrations [21]. In the standard setting, an expert shows behaviour by repeatedly interacting with the environment. This behaviour, encoded by its policy, is optimizing an *unknown* reward function. The goal of IRL consists of finding a reward function that makes the expert's behaviour optimal [20]. Compared to other imitation learning approaches [3, 15], which output an imitating policy (e.g, Behavioral Cloning [3]), IRL explicitly provides a succinct representation of the expert's *intention*. For this reason, it provides a generalization of the expert's policy to unobserved situations.

However, in some cases, it is not possible to wait for the convergence of the demonstrator's learning process. For instance, in multi-agent environments, an agent has to infer the unknown reward functions that the other agents are learning, before actually becoming "experts"; so that she can either cooperate or compete with them. On the other hand, in many situations, we can learn something useful by observing the learning process of an agent. These observations contain important information about the agent's intentions and can be used to infer her interests. Imagine a driver who is learning a new circuit. During her training, we can observe how she behaves in a variety of situations (even dangerous ones) and this is useful for understanding which states are good and which should be avoided. Instead, when expert behaviour is observed, only a small sub-region of the state space could be explored, thus leaving the observer unaware of what to do in situations that are unlikely under the expert policy.

Inverse Reinforcement Learning from not expert agents, called Learning from a Learner (LfL), was recently proposed by Jacq et al. in [17]. LfL involves two agents: a *learner* who is currently learning a task and an *observer* who wants to infer the learner's intentions. In [17] the authors assume that the learner is learning under an entropy-regularized framework, motivated by the assumption that the learner is showing a sequence of constantly improving policies. However many Reinforcement Learning (RL) algorithms [35] do not satisfy this and also human learning is characterized by mistakes that may lead to a non-monotonic learning process.

In this paper we propose a new algorithm for the LfL setting called Learning Observing a Gradient not-Expert Learner (LOGEL), which is not affected by the violation of the constantly improving assumption. Given that many successful RL algorithms are gradient-based [22] and there is some evidence that the human learning process is similar to a gradient-based method [32], we assume that the learner is following the gradient direction of her expected discounted return. The algorithm learns the reward function that minimizes the distance between the actual policy parameters of the learner and the policy parameters that should be obtained if she were following the policy gradient using that reward function.

After a formal introduction of the LfL setting in Section 3, we provide in Section 4 a first solution of the LfL problem when the observer has full access to the learner's policy parameters and learning rates. Then, in Section 5 we extend the algorithm to the more realistic case in which the observer can identify the optimized reward function only by analyzing the learner's trajectories. For each problem setting, we provide a finite sample analysis to give to the reader an intuition on the correctness of the recovered weights. Finally, we consider discrete and continuous simulated domains to empirically compare the proposed algorithm with state-of-the-art baselines in this setting [17, 7]. The proofs of all the results are reported in Appendix A. In the appendix we report preliminary results on a simulated autonomous driving task B.3.

## 2  Preliminaries

A **Markov Decision Process** (MDP) [27, 35] is a tuple $\mathcal{M} = (\mathcal{S}, \mathcal{A}, P, \gamma, \mu, R)$ where $\mathcal{S}$ is the state space, $\mathcal{A}$ is the action space, $P : \mathcal{S} \times \mathcal{A} \times \mathcal{S} \to \mathbb{R}_{\geq 0}$ is the transition function, which defines the density $P(s'|s,a)$ of state $s' \in \mathcal{S}$ when taking action $a \in \mathcal{A}$ in state $s \in \mathcal{S}$, $\gamma \in [0, 1)$ is the discount factor, $\mu : \mathcal{S} \to \mathbb{R}_{\geq 0}$ is the initial state distribution and $R : \mathcal{S} \to \mathbb{R}$ is the reward function. An RL agent follows a policy $\pi : \mathcal{S} \times \mathcal{A} \to \mathbb{R}_{\geq 0}$, where $\pi(\cdot|s)$ specifies for each state $s$ a distribution over the action space $\mathcal{A}$, i.e., the probability of taking action $a$ in state $s$. We consider stochastic differentiable policies belonging to a parametric space $\Pi_\Theta = \{\pi_{\boldsymbol{\theta}} : \boldsymbol{\theta} \in \Theta \subseteq \mathbb{R}^d\}$. We evaluate the performance of a policy $\pi_{\boldsymbol{\theta}}$ as its expected cumulative discounted return:

$$J(\boldsymbol{\theta}) = \mathop{\mathbb{E}}_{\substack{S_0 \sim \mu, \\ A_t \sim \pi_{\boldsymbol{\theta}}(\cdot|S_t), \\ S_{t+1} \sim P(\cdot|S_t, A_t)}} \left[ \sum_{t=0}^{+\infty} \gamma^t R(S_t, A_t) \right].$$

To solve an MDP, we must find a policy $\pi_{\boldsymbol{\theta}^*}$ that maximizes the performance $\boldsymbol{\theta}^* \in \arg\max_{\boldsymbol{\theta}} J(\boldsymbol{\theta})$.

**Inverse Reinforcement Learning** [21, 20, 2] addresses the problem of recovering the *unknown* reward function optimized by an expert given demonstrations of her behavior. The expert plays a policy $\pi^E$ which is (nearly) optimal for some unknown reward function $R : \mathcal{S} \times \mathcal{A} \to \mathbb{R}$. We are given a dataset $D = \{\tau_1, \ldots, \tau_n\}$ of trajectories from $\pi^E$, where we define a *trajectory* as a sequence of states and actions $\tau = (s_0, a_0, \ldots, s_{T-1}, a_{T-1}, s_T)$, where $T$ is the trajectory length. The goal of an IRL agent is to find a reward function that explains the expert's behavior. As commonly done in the Inverse Reinforcement Learning literature [24, 41, 2], we assume that the expert's reward function can be represented by a linear combination with weights $\boldsymbol{\omega}$ of $q$ basis functions $\boldsymbol{\phi}$:

$$R_{\boldsymbol{\omega}}(s, a) = \boldsymbol{\omega}^T \boldsymbol{\phi}(s, a), \quad \boldsymbol{\omega} \in \mathbb{R}^q, \tag{1}$$

where $\boldsymbol{\phi} : \mathcal{S} \times \mathcal{A} \to [-M_r, M_r]^q$ is a bounded feature vector function.

We define the **feature expectations** of a policy $\pi_{\boldsymbol{\theta}}$ as:

$$\boldsymbol{\psi}(\boldsymbol{\theta}) = \mathop{\mathbb{E}}_{\substack{S_0 \sim \mu, \\ A_t \sim \pi_{\boldsymbol{\theta}}(\cdot|S_t), \\ S_{t+1} \sim P(\cdot|S_t, A_t)}} \left[ \sum_{t=0}^{+\infty} \gamma^t \boldsymbol{\phi}(S_t, A_t) \right].$$

The **expected discounted return**, under the linear reward model, is defined as:

$$J(\boldsymbol{\theta}, \boldsymbol{\omega}) = \mathop{\mathbb{E}}_{\substack{S_0 \sim \mu, \\ A_t \sim \pi_{\boldsymbol{\theta}}(\cdot | S_t), \\ S_{t+1} \sim P(\cdot | S_t, A_t)}} \left[ \sum_{t=0}^{+\infty} \gamma^t R_{\boldsymbol{\omega}}(S_t, A_t) \right] = \boldsymbol{\omega}^T \boldsymbol{\psi}(\boldsymbol{\theta}). \tag{2}$$

## 3   Inverse Reinforcement Learning from learning agents

The Learning from a Learner Inverse Reinforcement Learning setting (LfL), proposed in [17], involves two agents:

- a *learner* which is learning a task defined by the reward function $R_{\boldsymbol{\omega}^L}$,
- and an *observer* which wants to infer the learner's reward function.

More formally, the learner is an RL agent which is learning a policy $\pi_{\boldsymbol{\theta}} \in \Pi_{\Theta}$ in order to maximize its *discounted expected return* $J(\boldsymbol{\theta}, \boldsymbol{\omega}^L)$. The learner is improving its own policy by an update function $f(\boldsymbol{\theta}, \boldsymbol{\omega}) : \mathbb{R}^d \times \mathbb{R}^q \to \mathbb{R}^d$, i.e., at time $t$, $\boldsymbol{\theta}_{t+1} = f(\boldsymbol{\theta}_t, \boldsymbol{\omega})$. The observer, instead, perceives a sequence of learner's policy parameters $\{\boldsymbol{\theta}_1, \cdots, \boldsymbol{\theta}_{m+1}\}$ and a dataset of trajectories for each policy $\mathcal{D} = \{\mathcal{D}_1, \cdots, \mathcal{D}_{m+1}\}$, where $\mathcal{D}_i = \{\tau_1^i, \cdots, \tau_n^i\}$. Her goal is to recover the reward function $R_{\boldsymbol{\omega}^L}$ that explains $\pi_{\boldsymbol{\theta}_i} \to \pi_{\boldsymbol{\theta}_{i+1}}$ for all $1 \leq i \leq m$, i.e the updates of the learner's policy.

**Remark 3.1.** *It is easy to notice that this problem has the same intention as Inverse Reinforcement Learning since the demonstrating agent is motivated by some reward function. On the other hand, in classical IRL the learner agent is an expert, and not a non-stationary agent. For this reason, we cannot simply apply standard IRL algorithms to this problem or use Behavioral Cloning [26, 3, 21] algorithms, which mimic a suboptimal behavior.*

## 4   Learning from a learner following the gradient

Many algorithms that are the state of the art of reinforcement learning are policy-gradient methods [22, 36], i.e. approaches which optimize the expected discounted return with gradient updates of the policy parameters. Recently it has been proved that even standard RL algorithms such as Value Iteration or Q-learning have strict connections with policy gradient methods [13, 30]. For the above reasons, we assume that the learner is optimizing the expected discounted return using gradient descent.

For the sake of presentation, we start by considering the simplified case in which we assume that the observer can perceive the sequence of the learner's policy parameters $(\boldsymbol{\theta}_1, \cdots, \boldsymbol{\theta}_{m+1})$, the associated gradients of the feature expectations $(\nabla_{\boldsymbol{\theta}} \boldsymbol{\psi}(\boldsymbol{\theta}_1), \ldots, \nabla_{\boldsymbol{\theta}} \boldsymbol{\psi}(\boldsymbol{\theta}_m))$, and the learning rates $(\alpha_1, \cdots, \alpha_m)$. Then, we will replace the exact knowledge of the gradients with estimates built on a set of demonstrations $\mathcal{D}_i$ for each learner's policy $\pi_{\boldsymbol{\theta}_i}$ (Section 4.2). Finally, we introduce our algorithm LOGEL, which, using behavioral cloning and an alternate block-coordinate optimization [37], is able to estimate the reward's parameters without requiring as input the policy parameters and the learning rates (Section 5).

### 4.1   Exact gradient

We express the gradient of the expected return as [36, 23]:

$$\nabla_{\boldsymbol{\theta}} J(\boldsymbol{\theta}, \boldsymbol{\omega}) = \mathop{\mathbb{E}}_{\substack{S_0 \sim \mu, \\ A_t \sim \pi_{\boldsymbol{\theta}}(\cdot | S_t), \\ S_{t+1} \sim P(\cdot | S_t, A_t)}} \left[ \sum_{t=0}^{+\infty} \gamma^t R_{\boldsymbol{\omega}}(S_t, A_t) \sum_{l=0}^{t} \nabla_{\boldsymbol{\theta}} \log \pi_{\boldsymbol{\theta}}(A_l | S_l) \right] = \nabla_{\boldsymbol{\theta}} \boldsymbol{\psi}(\boldsymbol{\theta}) \boldsymbol{\omega},$$

where $\nabla_{\boldsymbol{\theta}} \boldsymbol{\psi}(\boldsymbol{\theta}) = (\nabla_{\boldsymbol{\theta}} \psi_1(\boldsymbol{\theta}) | \ldots | \nabla_{\boldsymbol{\theta}} \psi_q(\boldsymbol{\theta})) \in \mathbb{R}^{d \times q}$ is the Jacobian matrix of the feature expectations $\boldsymbol{\psi}(\boldsymbol{\theta})$ w.r.t the policy parameters $\boldsymbol{\theta}$. In the rest of the paper, with some abuse of notation, we will indicate $\boldsymbol{\psi}(\boldsymbol{\theta}_t)$ with $\boldsymbol{\psi}_t$.

We define the gradient-based learner updating rule at time $t$ as:

$$\boldsymbol{\theta}_{t+1}^L = \boldsymbol{\theta}_t^L + \alpha_t \nabla_{\boldsymbol{\theta}} J(\boldsymbol{\theta}_t^L, \boldsymbol{\omega}) = \boldsymbol{\theta}_t^L + \alpha_t \nabla_{\boldsymbol{\theta}} \boldsymbol{\psi}_t^L \boldsymbol{\omega}^L, \tag{3}$$

where $\alpha_t$ is the learning rate. Given a sequence of consecutive policy parameters $(\boldsymbol{\theta}_1^L, \cdots, \boldsymbol{\theta}_{m+1}^L)$, and of learning rates $(\alpha_1, \cdots, \alpha_m)$ the observer has to find the reward function $R_{\boldsymbol{\omega}}$ such that the improvements are explainable by the update rule in Eq. (3). This implies that the observer has to solve the following minimization problem:

$$\min_{\boldsymbol{\omega} \in \mathbb{R}^q} \sum_{t=1}^{m} \| \Delta_t - \alpha_t \nabla_{\boldsymbol{\theta}} \boldsymbol{\psi}_t \boldsymbol{\omega} \|_2^2, \tag{4}$$

where $\Delta_t = \boldsymbol{\theta}_{t+1} - \boldsymbol{\theta}_t$. This optimization problem can be easily solved in closed form under the assumption that $\left( \sum_{t=1}^{m} \alpha_t^2 \nabla_{\boldsymbol{\theta}} \boldsymbol{\psi}_t^T \nabla_{\boldsymbol{\theta}} \boldsymbol{\psi}_t \right)^{-1}$ is invertible.

**Lemma 4.1.** *If the matrix* $\left( \sum_{t=1}^{m} \alpha_t^2 \nabla_{\boldsymbol{\theta}} \boldsymbol{\psi}_t^T \nabla_{\boldsymbol{\theta}} \boldsymbol{\psi}_t \right)^{-1}$ *is full-rank than optimization problem* (4) *is solved in closed form by*

$$\widehat{\boldsymbol{\omega}} = \left( \sum_{t=1}^{m} \alpha_t^2 \nabla_{\boldsymbol{\theta}} \boldsymbol{\psi}_t^T \nabla_{\boldsymbol{\theta}} \boldsymbol{\psi}_t \right)^{-1} \left( \sum_{t=1}^{m} \alpha_t \nabla_{\boldsymbol{\theta}} \boldsymbol{\psi}_t^T \Delta_t \right). \tag{5}$$

When problem (4) has no unique solution or when the matrix to be inverted is nearly singular, in order to avoid numerical issues, we can resort to a regularized version of the optimization problem. In the case we add an L2-norm penalty term over weights $\boldsymbol{\omega}$ we can still compute a closed-form solution (see Lemma A.5 in Appendix A).

## 4.2 Approximate gradient

In practice, we do not have access to the Jacobian matrix $\nabla_{\boldsymbol{\theta}} \boldsymbol{\psi}$, but the observer has to estimate it using the dataset $D$ and some unbiased policy gradient estimator, such as REINFORCE [40] or G(PO)MDP [5]. The estimation of the Jacobian will introduce errors on the optimization problem (4). Obviously the estimation of the reward weights $\boldsymbol{\omega}$ becomes more accurate when more data are available [25]. On the other hand during the learning process, the learner will produce more than one policy improvement, and the observer can use these improvements to get better estimates of the reward weights.

In order to have an insight on the relationship between the amount of data needed to estimate the gradient and the number of learning steps, we provide a finite sample analysis on the norm of the difference between the learner's weights $\boldsymbol{\omega}^L$ and the recovered weights $\widehat{\boldsymbol{\omega}}$. The analysis takes into account the learning steps data and the gradient estimation data, without having any assumption on the policy of the learner. We denote with $\boldsymbol{\Psi} = [\nabla_{\boldsymbol{\theta}} \boldsymbol{\psi}_1, \cdots, \nabla_{\boldsymbol{\theta}} \boldsymbol{\psi}_m]^T$ the concatenation of the Jacobians and $\widehat{\boldsymbol{\Psi}} = \left[ \widehat{\nabla_{\boldsymbol{\theta}} \boldsymbol{\psi}}_1, \cdots, \widehat{\nabla_{\boldsymbol{\theta}} \boldsymbol{\psi}}_m \right]^T$ the concatenation of the estimated Jacobians.

**Theorem 4.1.** *Let* $\boldsymbol{\Psi}$ *be the real Jacobians and* $\widehat{\boldsymbol{\Psi}}$ *the estimated Jacocobian from* $n$ *trajectories* $\{\tau_1, \cdots, \tau_n\}$. *Assume that* $\boldsymbol{\Psi}$ *is bounded by a constant* $M$ *and* $\lambda_{\min}(\widehat{\boldsymbol{\Psi}}^T \widehat{\boldsymbol{\Psi}}) \geq \lambda > 0$. *Then w.h.p.:*

$$\left\| \boldsymbol{\omega}^L - \widehat{\boldsymbol{\omega}} \right\|_2 \leq O \left( \frac{1}{\lambda} M \sqrt{\frac{dq \log(\frac{2}{\delta})}{2n}} \left( \sqrt{\frac{\log dq}{m}} + \sqrt{dq} \right) \right).$$

We have to underline that a finite sample analysis is quite important for this problem. In fact, the number of policy improvement steps of the learner is finite as the learner will eventually achieve an optimal policy. So, knowing the finite number of learning improvements $m$, we can estimate how much data we need for each policy to get an estimate with a certain accuracy. More information about the proof of the theorem can be found in appendix A.

**Remark 4.1.** *Another important aspect to take into account is that there is an intrinsic bias [18] due to the gradient estimation error that cannot be solved by increasing the number of learning steps, but only with a more accurate estimation of the gradient. However, we show in Section 7 that, experimentally, the component of the bound that does not depend on the number of learning steps does not influence the recovered weights.*

# 5 Learning from improvement trajectories

In a realistic scenario, the observer has access only to a dataset $\mathcal{D} = (\mathcal{D}_1, \ldots, \mathcal{D}_{m+1})$ of trajectories generated by each policy, such that $\mathcal{D}_i = \{\tau_1, \cdots, \tau_n\} \sim \pi_{\boldsymbol{\theta}_i}$. Furthermore, the learning rates are unknown and possibly the learner applies an update rule other than (3). The observer has to infer the policy parameters $\Theta = (\boldsymbol{\theta}_1, \ldots, \boldsymbol{\theta}_{m+1})$, the learning rates $A = (\alpha_1, \ldots, \alpha_m)$, and the reward weights $\boldsymbol{\omega}$. If we suppose that the learner is updating its policy parameters with gradient ascent on the discounted expected return, the natural way to see this problem is to maximize the log-likelihood of $p(\boldsymbol{\theta}_1, \boldsymbol{\omega}, A|D)$:

$$\max_{\boldsymbol{\theta}_1, \boldsymbol{\omega}, A} \sum_{(s,a) \in \mathcal{D}_1} \log \pi_{\boldsymbol{\theta}_1}(a|s) + \sum_{t=2}^{m+1} \sum_{(s,a) \in \mathcal{D}_t} \log \pi_{\boldsymbol{\theta}_t}(a|s),$$

where $\boldsymbol{\theta}_t = \boldsymbol{\theta}_{t-1} + \alpha_{t-1} \nabla_{\boldsymbol{\theta}} \psi_{t-1}$. Unfortunately, solving this problem directly is not practical as it involves evaluating gradients of the discounted expected return up to the $m$-th order. To deal with this, we break down the inference problem into two steps: the first one consists in recovering the policy parameters $\Theta$ of the learner and the second in estimating the learning rates $A$ and the reward weights $\boldsymbol{\omega}$ (see Algorithm 1).

## 5.1 Recovering learner policies

Since we assume that the learner's policy belongs to a parametric policy space $\Pi_\Theta$ made of differentiable policies, as explained in [24], we can recover an approximation of the learner's parameters $\Theta$ through behavioural cloning, exploiting the trajectories in $\mathcal{D} = \{\mathcal{D}_1, \cdots, \mathcal{D}_{m+1}\}$. For each dataset $\mathcal{D}_i \in \mathcal{D}$ of trajectories, we cast the problem of finding the parameter $\boldsymbol{\theta}_i$ to a maximum-likelihood estimation. Solving the following optimization problem we obtain an estimate $\widehat{\boldsymbol{\theta}}_i$ of $\boldsymbol{\theta}_i$:

$$\max_{\boldsymbol{\theta}_i \in \Theta} \frac{1}{n} \sum_{l=1}^{n} \sum_{t=0}^{T-1} \log \pi_{\boldsymbol{\theta}_i}(a_{l,t}|s_{l,t}). \tag{6}$$

It is known that the maximum-likelihood estimator is consistent under mild regularity conditions on the policy space $\Pi_\Theta$ and assuming the identifiability property [8]. Some finite-sample guarantees on the concentration of distance $\|\widehat{\boldsymbol{\theta}}_i - \boldsymbol{\theta}_i\|_p$ were also derived under stronger assumptions, e.g., in [34].

## 5.2 Recovering learning rates and reward weights

Given the parameters $(\widehat{\boldsymbol{\theta}}_1, \ldots, \widehat{\boldsymbol{\theta}}_{m+1})$, if the learner is updating her policy with a constant learning rate we can simply apply Eq. (4). On the other hand, with an unknown learner, we cannot make this assumption and it is necessary to estimate also the learning rates $A = (\alpha_1, \ldots, \alpha_m)$. The optimization problem in Eq. (4) becomes:

$$\min_{\boldsymbol{\omega} \in \mathbb{R}^q, A \in \mathbb{R}^m} \sum_{t=1}^{m} \left\| \widehat{\Delta}_t - \alpha_t \widehat{\nabla_{\boldsymbol{\theta}} \psi_t} \boldsymbol{\omega} \right\|_2^2 \tag{7}$$

$$\text{s.t.} \quad \alpha_t \geq \epsilon \quad 1 \leq t \leq m. \tag{8}$$

where $\widehat{\Delta}_t = \widehat{\boldsymbol{\theta}}_{t+1} - \widehat{\boldsymbol{\theta}}_t$ and $\epsilon$ is a small constant. To optimize this function we use alternate block-coordinate descent [37]. We alternate the optimization of parameters $A$ and the optimization of parameters $\boldsymbol{\omega}$. Furthermore, we can notice that these two steps can be solved in closed form. When we optimize on $\boldsymbol{\omega}$, the optimization can be done using Lemma 4.1. When we optimize on $A$ we can solve for each parameter $\alpha_t$, with $1 \leq t \leq m$, in closed form.

**Lemma 5.1.** *The minimum of (7) with respect to $\alpha_t$ is equal to:*

$$\hat{\alpha}_t = \max \left( \epsilon, \left( (\widehat{\nabla_{\boldsymbol{\theta}} \psi_t} \boldsymbol{\omega})^T (\widehat{\nabla_{\boldsymbol{\theta}} \psi_t} \boldsymbol{\omega}) \right)^{-1} (\widehat{\nabla_{\boldsymbol{\theta}} \psi_t} \boldsymbol{\omega})^T \hat{\Delta}_t \right). \tag{9}$$

The inner matrix cannot be inverted only if the vector $\widehat{\nabla_{\boldsymbol{\theta}} \psi} \boldsymbol{\omega}$ is equal to $\mathbf{0}$. This would happen only if the expert is at a stationary point, so $\widehat{\nabla_{\boldsymbol{\theta}} \psi}$ is $\mathbf{0}$. The optimization converges under the assumption that there exists a unique minimum for each variable $A$ and $\boldsymbol{\omega}$ [37].

**Algorithm 1** LOGEL

**Require:** Dataset $\mathcal{D} = \{\mathcal{D}_1, \ldots, \mathcal{D}_{m+1}\}$ with $\mathcal{D}_j = \{(\tau_1, \ldots, \tau_{n_j}) \mid \tau_i \sim \pi_{\boldsymbol{\theta}_j}\}$
**Ensure:** Reward weights $\boldsymbol{\omega} \in \mathbb{R}^q$

1: Estimate policy parameters $(\hat{\boldsymbol{\theta}}_1, \ldots, \hat{\boldsymbol{\theta}}_{m+1})$ with Eq. (6)
2: Initialize $A$ and $\boldsymbol{\omega}$
3: Compute learning rates $A$ and reward weights $\boldsymbol{\omega}$ by alternating (9) and (5) up to convergence

## 5.3 Theoretical analysis

In this section, we provide a finite-sample analysis of LOGEL when only one learning step is observed, assuming that the Jacobian matrix $\widehat{\nabla_{\boldsymbol{\theta}}\psi}$ is bounded and the learner's policy is a Gaussian policy $\pi \sim \mathcal{N}(\boldsymbol{\theta}^T \boldsymbol{\varphi}(s), \sigma^2)$. The analysis evaluates the norm of the difference between the learner's weights $\boldsymbol{\omega}^L$ and the recovered weights $\hat{\boldsymbol{\omega}}$. Without loss of generality, we consider the case where the learning rate $\alpha = 1$. The analysis takes into account the bias introduced by the behavioural cloning and the gradient estimation.

**Theorem 5.1.** *Let $\pi_{\boldsymbol{\theta}_1}, \pi_{\boldsymbol{\theta}_2}$ be two Gaussian policies $\pi_{\boldsymbol{\theta}_i}(\cdot|s) \sim \mathcal{N}(\boldsymbol{\theta}_i^T \boldsymbol{\varphi}(s), \sigma^2)$ with $i \in \{1, 2\}$, such that $\pi_{\boldsymbol{\theta}_2}$ is the improvement of $\pi_{\boldsymbol{\theta}_1}$. Let $i \in [1, 2]$. Given datasets $D_i = \{\tau_1^i, \ldots, \tau_n^i\}$ of trajectories generated by $\pi_i$, such that $S_i \in \mathbb{R}^{n \times t \times d}$ is the matrix of corresponding states features, let the minimum singular value of $\sigma_{\min}(S_i^T S_i) \geq \eta > 0$, $\widehat{\nabla_{\boldsymbol{\theta}_1}\psi}$ uniformly bounded by $M$, the state features bounded by $M_S$, and the reward features bounded by $M_R$. Then with probability $1 - \delta$:*

$$\left\| \boldsymbol{\omega}^L - \widehat{\boldsymbol{\omega}} \right\|_2 \leq O\left( \frac{(M + M_S^2 M_R)}{\sigma_{\min}(\nabla_{\boldsymbol{\theta}_1}\psi)} \sqrt{\frac{\log(\frac{2}{\delta})}{n\eta}} \right)$$

*where $\omega^L$ are the real reward parameters and $\widehat{\omega}$ are the parameters recovered using Lemma 4.1.*

The theorem, that relies on perturbation analysis [39] and least squares with fixed design [29], underlines how LOGEL, with a sufficient number of samples to estimate the policy parameters and the gradients, succeeds in recovering the correct reward parameters.

## 6 Related Works

The problem of estimating the reward function of an agent who is learning is quite new. This setting was proposed by Jacq et al. [17] and, to the best of our knowledge, it is studied only in that work. In [17] the authors proposed a method based on entropy-regularized reinforcement learning, in which they assumed that the learner is performing soft policy improvements. In order to derive their algorithm, the authors also assume that the learner respects the policy improvement condition. We do not make this assumption as our formulation assumes only that the learner is changing its policy parameters along the gradient direction (which can result in a performance loss).

The problem, as we underlined in Section 3, is close to the Inverse Reinforcement Learning problem [21, 20], since they share the intention of acquiring the unknown reward function from the observation of an agent's demonstrations. LOGEL relies on the assumption that the learner is improving her policy through gradient ascent updates. A similar approach, but in the expert case, was taken in [25, 19] where the authors use the null gradient assumption to learn the reward from expert's demonstrations.

In another line of works, sub-optimal demonstrations are used in the preference-based IRL [11, 16] and ranking-based IRL [7, 9]. Some of these works require that the algorithm asks a human to compare possible agent's trajectories to learn the underlying reward function of the task. We can imagine that LOGEL can be used in a similar way to learn from humans who are learning a new task. Instead, in [4] was proposed an Imitation Learning setting where the observer tries to imitate the behavior of a supervisor that demonstrates a converging sequence of policies.

In works on *theory of minds* [28, 33], the authors propose an algorithm that uses meta-learning to build a system that learns how to model other agents. In these works it is not required that agents are experts but they must be stationary. Instead, in the setting considered by LOGEL, the observed agent is non-stationary.

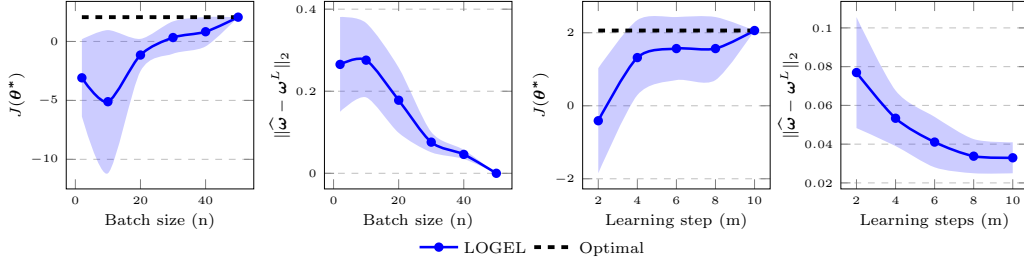

Figure 1: Gridworld experiment with known policy parameters. The learner is using G(PO)MDP. From left the expected discounted return and the norm difference between the real weights and the recovered ones with one learning step; the same measures with fixed batch size (5 trajectories with length 20). The performance of the observers are evaluated on the learner's reward weights. Results are averaged over 20 runs. $98\%$ c.i as shaded area.

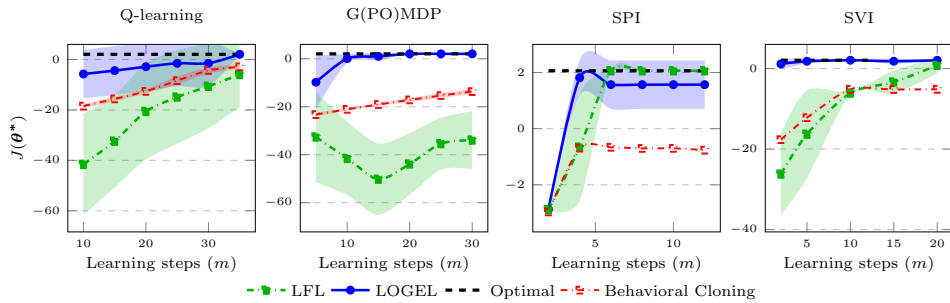

Figure 3: Gridworld experiment with estimated policy parameters and four learners: from left Q-learning, G(PO)MDP, SPI, SVI. The green line is the LfL observer, the blue one is the LOGEL observer and the red one Behavioral Cloning. The performance of the observers are evaluated on the learner's reward weights. Results are everaged over 20 runs. $98\%$ c.i. as shaded area.

# 7 Experiments

This section is devoted to the experimental evaluation of LOGEL. The algorithm LOGEL is compared to the state-of-the-art baseline Learner From a Learner (LfL) [17] and T-REX [7] in a gridworld navigation task and in two MuJoCo environments. In these experiments the assumption that the learner is gradient-based is violated and in the MuJoCo task the reward features are constructed by states and actions features. Therefore we can argue that in this experiment the reward linearity assumption is violated, since we use a different reward space for the recovered reward function. More details on the experiments are in Appendix B.1.

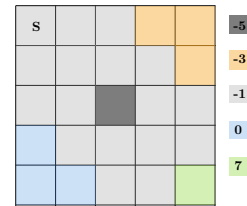

Figure 2: Gridworld environment: every area has a different reward weight. In the green area the agent is reset to the starting state.

## 7.1 Gridworld

The first set of experiments aims at evaluating the performance of LO-GEL in a discrete Gridworld environment. The Gridworld, represented in Figure 2, is composed of five regions with a different reward for each area. The agent starts from the cell top left and when she reaches the green state, then returns to the starting state. The reward feature space is composed of the one-hot encoding of five areas: the orange, the light grey, the dark grey, the blue, and the green. The learner weights for the areas are $(-3, -1, -5, 7, 0)$ respectively. As a first experiment, we want to verify in practice the theoretical finding exposed in Section 4. In this experiment, the learner uses a Boltzmann policy and she is learning with the G(PO)MDP policy gradient algorithm. The observer has access to the true policy parameters of the learner. Figure 1 shows the performance of LOGEL in two settings: a single learning step and increasing batch size (5, 10, 20, 30, 40, 50); a fixed batch size (batch size 5 and trajectory length 20) and an increasing number

of learning steps (2, 4, 6, 8, 10). The figure shows the expected discounted return (evaluated in closed form) and the difference in norm between the learner's weights and the recovered weights [1]. We note that, as explained in Theorem 4.1, with a more accurate gradient estimate, the observer succeeds in recovering the reward weights by observing even just one learning step. On the other hand, as we can deduce from Theorem 4.1, if we have a noisy estimation of the gradient, with multiple learning steps, the observer succeeds in recovering the learner's weights. It is interesting to notice that, from this experiment, it seems that the bias component, which does not vanish as the learning steps increase (see Theorem 4.1), does not affect the correctness of the recovered weights.

In the second experiment we consider four different learners using: Q-learning [35], G(PO)MDP, Soft policy improvement (SPI) [17] and Soft Value Iteration (SVI) [14] [2]. For this experiment, we compare the performance of LOGEL , LfL [17] and Behavioral Cloning. In Figure 3 we can notice as LOGEL succeeds in recovering the learner's reward weights even with learner algorithms other than gradient-based ones. Instead, LfL does not recover the reward weights of the G(PO)MDP learner and needs more learning steps than LOGEL to learn the reward weights when Q-learning learner and SVI are observed. Behavioral Cloning only mimics the last seen policy which can be suboptimal.

## 7.2 MuJoCo environments

In the second set of experiments, we show the ability of LOGEL to infer the reward weights in more complex and continuous environments. We use two environments from the MuJoCo control suite [6]: Hopper and Reacher. As in [17], the learner is trained using Policy Proximal Optimization (PPO) [31][3], with 16 parallel agents for each learning step. For each step, the length of the trajectories is 2000. Then we use LOGEL , LfL or T-REX [7] to recover the reward parameters. In the end, the observer is trained with the recovered weights using PPO and the performances are evaluated on the learner's weights, starting from the same initial policy of the learner for a fair comparison. The scores are normalized by setting to 1 the score of the last observed learner policy and to 0 the score of the initial one (as done in [17]). In both environments, the observer learns using the learning steps from 10 to 20 as the first learning steps are too noisy. The reward function of LfL is the same as the one used in the original paper, where the reward function is a neural network equal to the one used for the learner's policy.

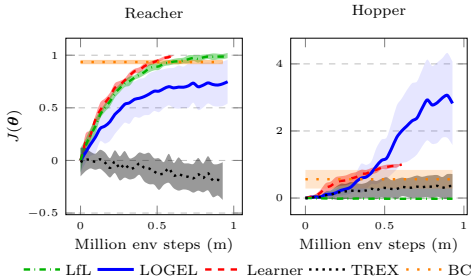

Figure 4: From the left, the Reacher and the Hopper MuJoCo environments. The red line is the performance of the learner during 20 learning steps. The observers, LfL, LOGEL, T-REX and Behavioral Cloning (BC) observe the trajectories of the learning steps from 10 to 20. The performance of the observers are evaluated on the learner's reward weights. Scores are normalized setting to 0 the first return of the learner and to 1 the last one. The results are averaged over 10 runs. 98% c.i. are shown as shaded areas.

Instead, for LOGEL we used linear reward functions derived only from state and action features. The reward for the Reacher environment is a 26-grid radial basis function that describes the distance between the agent and the goal, plus the 2-norm squared of the action. In the Hopper environment, instead, the reward features are the distance between the previous and the current position and the 2-norm squared of the action. The T-REX algorithm aims to recover a reward function from ranked trajectories, where the rank is given by an oracle and is based on the expected discounted return. We use the algorithm in the LfL setting, where we approximate the ranking with the temporal updates of the policies, as was done in an example in the original paper. We implement the reward function as in the original paper with a three layer neural network with 256 as hidden size.

The results are shown in Figure 4, where we reported results averaged over 10 runs. For Behavioral Cloning we report the performance of the policy learnt at the 20th step; in fact Behavioral Cloning cannot improve its performance with learning steps. We can notice that LOGEL succeeds in

identifying a good reward function in both environments, although in the Reacher environment the recovered reward function causes slower learning. Instead, LfL fails to recover an effective reward function for the Hopper environment [17]. The T-REX algorithm, as in the original paper, succeeds in recovering a good approximation of the reward weights in the Hopper domain; instead, it does not succeed into recovering the reward function of the Reacher environment.

# 8    Conclusions

In this paper we propose a novel algorithm, LOGEL, for the "Learning from a Learner Inverse Reinforcement Learning" setting. The proposed method relies on the assumption that the learner updates her policy along the direction of the gradient of the expected discounted return. We provide some finite-sample bounds on the algorithm performance in recovering the reward weights when the observer observes the learner's policy parameters and when the observer observes only the learner's trajectories. Finally, we tested LOGEL on a discrete gridworld environment and on two MuJoCo continuous environments, comparing the algorithm with the state-of-the-art baselines [17, 7]. As future work, we plan to extend the algorithm to account for the uncertainty in estimating both the policy parameters and the gradient.

## Broader impact

In this paper, we focus on the Inverse Reinforcement Learning [2, 15, 3, 21] task from a Learning Agent [17]. The first motivation to study Inverse Reinforcement Learning algorithms is to overcome the difficulties that can arise in specifying the reward function from human and animal behaviour. Sometimes, in fact, it is easier to infer human intentions by observing their behaviours than to design a reward function by hand. An example is helicopter flight control [1], in which we can observe a helicopter operator and through IRL a reward function is inferred to teach a physical remote-controlled helicopter. Another example is to predict the behavior of a real agent as route prediction tasks of taxis [41, 42] or anticipation of pedestrian interactions [12] or energy-efficient driving [38]. However, in many cases, the agents are not really experts and on the other hand, only expert demonstrations can not show their intention to avoid dangerous situations. We want to point out that learning what the agent wants to avoid because harmful is as important as learning his intentions.

The possible outcomes of this research are the same as those of Inverse Reinforcement Learning mentioned above, avoiding the constraint that the agent has to be an expert. In future work, we will study how to apply the proposed algorithm in order to infer the pilot's intentions when they learn a new circuit.

A relevant possible complication of using IRL is the error on the reward feature engineering which can lead to errors in understanding the agent's intentions. In an application such as autonomous driving, errors in the reward function can cause dangerous situations. For this reason, verification through the simulated environment of the effectiveness of the retrieve rewards is quite important.

## Funding Transparency Statement

This work has been partially supported by the Italian MIUR PRIN 2017 Project ALGADIMAR "Algorithms, Games, and Digital Market".

## Footnotes

[1]To perform this comparison, we normalize the recovered weights and the learner's weights

[2]In Appendix B.1 the learning process of each learning agent is shown.

[3]It is important to notice that PPO violates the gradient learning assumption of LOGEL.

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
