[Supplementary Material]

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

# A  Proofs and derivations

We start the proofs given some introduction on Pertubation on Least Square problems and on Least Square problems with fixed design. Then we report the proofs and derivations for the results of Sections 4 and 5. For the rest of this section we assume that:

- $\boldsymbol{\omega}^L, \widehat{\boldsymbol{\omega}} \in \mathbb{R}^q$,
- $\boldsymbol{\theta}^L, \widehat{\boldsymbol{\theta}} \in \mathbb{R}^d$.

We define with $\boldsymbol{\omega}^L$ ($\boldsymbol{\theta}^L$) the reward (policy) parameters of the learner, and with $\widehat{\boldsymbol{\omega}}$ ($\widehat{\boldsymbol{\theta}}$) the reward (policy) parameters recovered by the observer.

## A.1  Preliminaries

**Definition A.1.** *The condition number of a matrix $\mathbf{A} \in R^{m \times q}$ $\mathbf{A} \neq 0$ is:*

$$\kappa = \|\mathbf{A}\|_2 \, \|\mathbf{A}^+\|_2 = \frac{\sigma_1}{\sigma_r},$$

*where $0 < r = rank(\mathbf{A}) \leq \min(m, q)$, and $\sigma_1 \geq \cdots \geq \sigma_r > 0$ are the nonzero singular values of $\mathbf{A}$.*

A least-squares problem is defined as:

$$\min_x \|\mathbf{A}x - b\|_2, \tag{10}$$

where the solution is $x = \mathbf{A}^+ b$. We denote with $\mathbf{A}^+$ the pseudoinverse of $\mathbf{A}$, the perturbed $\mathbf{A}$ as $\widehat{\mathbf{A}} = \mathbf{A} + \delta\mathbf{A}$ and the pertubed $\hat{b} = b + \delta b$ and the perturbed solution $\widehat{x} = \widehat{\mathbf{A}}^+ \hat{b} = x + \delta x$. Finally, we denote with $\mathbf{A}^H$ the adjoint of the matrix $\mathbf{A}$.

We define as $\chi = \frac{\|\delta\mathbf{A}\|_2}{\|\mathbf{A}\|_2}$ and $y = \mathbf{A}^{+H} x$.

**Lemma A.1** (Perturbation on Least Square Problems [39])**.** *Assume that $rank(\mathbf{A} + \delta\mathbf{A}) = rank(\mathbf{A})$ and $\chi\kappa < 1$ then:*

$$\|x - \hat{x}\|_2 \leq \frac{\kappa}{(1 - \chi\kappa)\|\mathbf{A}\|_2} (\chi \|x\|_2 \|\mathbf{A}\|_2 + \chi\kappa \|r\|_2 + \|\delta b\|_2) + \chi \|y\|_2 \|\mathbf{A}\|_2. \tag{11}$$

*Proof.* The proof can be find in [39]. $\qquad\square$

We adapt the lemma 6 in [10] to our context where $\widehat{\boldsymbol{\omega}}$ are the reward weights recovered with lemma 4.1.

**Lemma A.2** (From lemma 6 in [10])**.** *Let $\Sigma = (\widehat{\nabla_{\boldsymbol{\theta}}\psi}^T \widehat{\nabla_{\boldsymbol{\theta}}\psi})$ and suppose the following strong convexity condition holds: $\lambda_{\min}(\Sigma) \geq \lambda > 0$. Then the estimation error satisfies:*

$$\left\|\widehat{\boldsymbol{\omega}} - \boldsymbol{\omega}^L\right\|_2 \leq O\left(\frac{1}{\lambda} \left\|\nabla_{\boldsymbol{\theta}}\psi^T \Delta - \Sigma\boldsymbol{\omega}^L\right\|_2\right).$$

**Lemma A.3** (Revised from lemma 11 in [18])**.** *Suppose $X \in \mathbb{R}^{m \times q}$ and $W \in \mathbb{R}^{n \times M}$ are zero-mean sub-gaussian matrices with parameters $(\frac{1}{n}\Sigma_x, \frac{1}{n}\sigma_x^2), (\frac{1}{n}\Sigma_w, \frac{1}{n}\sigma_w^2)$ respectively. Then for any fixed vectors $v_1, v_2$, we have:*

$$P[|v_1^T(W^T X - \mathbb{E}[W^T W])v_2| \geq t \|v_1\|_2 \|v_2\|_2] \leq 3 exp\left(-cn \min\left\{\frac{t^2}{\sigma_x^2\sigma_w^2}, \frac{t}{\sigma_x\sigma_w}\right\}\right),$$

*in particular if $n \gtrsim \log p$ we have that:*

$$|v_1^T(W^T X - \mathbb{E}[W^T W])v_2| \leq \sigma_x\sigma_w \|v_1\|_2 \|v_2\|_2 \sqrt{\frac{\log p}{n}}.$$

*Setting $v_1$ to be the first standard basis vector and using a union bound over $j = 1, \cdots, p$ we have:*

$$\left\|(W^T X - \mathbb{E}[W^T X])v\right\|_\infty \leq \sigma_x\sigma_w \|v\|_2 \sqrt{\frac{\log p}{n}},$$

*with probability $1 - c_1 exp(-c_2 \log p)$ where $c_1, c_2$ are positive constants which are independent from $\sigma_x, \sigma_w, n, p$.*

**Theorem A.1** (from Chapter 2 [29])**.** *Assume that the least-squares model:*

$$\min_x \|\mathbf{A}x - b + \epsilon\|$$

*holds where $\epsilon \sim subGn(\sigma^2)$. Then, for any $\delta > 0$, with probability $1 - \delta$ it holds:*

$$\|x - \hat{x}\|_2 \leq \sigma \sqrt{\frac{r + \log(\frac{1}{\delta})}{n\sigma_{\min}}},$$

*where $\sigma_{\min} = \frac{\mathbf{A}^T\mathbf{A}}{n}$ is the minimum singular value of $\mathbf{A}^T\mathbf{A}$ and $r$ is the rank($\mathbf{A}^T\mathbf{A}$).*

## A.2   Additional Results

In this section we give the proofs and derivations of the theorems in Section 4.

First, we will provide a finite sample analysis on the difference in norm between the reward vector of the learner $\boldsymbol{\omega}^L$ and the reward vector recoverd using (5), with a single learning step. This result was omitted in the main paper as we can see this as a special case of Theorem 4.1, but with a different technique. We add it here as it provides a first insight on how, having enough demonstrations, we can recover the correct weights. In the demonstration, without loss of generality, we assume that the learning rate is 1.

**Lemma A.4.** *Let $\nabla_{\boldsymbol{\theta}}\psi$ be the real Jacobian and $\widehat{\nabla_{\boldsymbol{\theta}}\psi}$ the estimated Jacobian from $n$ trajectories $\{\tau_1, \cdots, \tau_n\}$. Assume that $\widehat{\nabla_{\boldsymbol{\theta}}\psi}$ is uniformly bounded by $M$. Then with probability $1 - \delta$*

$$\left\|\widehat{\nabla_{\boldsymbol{\theta}}\psi} - \nabla_{\boldsymbol{\theta}}\psi\right\|_2 \leq M\sqrt{qd}\sqrt{\frac{\log(\frac{2}{\delta})}{2n}}.$$

*Proof.* We use Hoeffding's inequality:

$$\mathrm{P}\left[\left\|\widehat{\nabla_{\boldsymbol{\theta}}\psi} - \nabla_{\boldsymbol{\theta}}\psi\right\|_2 \geq t\right] \leq \mathrm{P}\left[\sqrt{qd}\left\|\widehat{\nabla_{\boldsymbol{\theta}}\psi} - \nabla_{\boldsymbol{\theta}}\psi\right\|_\infty \geq t\right] \leq 2\exp\left(\frac{-2t^2n}{dqM^2}\right)$$

The result follows by setting $\delta = 2\exp\left(\frac{-2t^2n}{dqM^2}\right)$.  □

**Theorem A.1.** *Let $\nabla_{\boldsymbol{\theta}}\psi$ be the real Jacobian and $\widehat{\nabla_{\boldsymbol{\theta}}\psi}$ the estimated Jacobian from $n$ trajectories $\{\tau_1, \cdots, \tau_n\}$. Assume that $\widehat{\nabla_{\boldsymbol{\theta}}\psi}$ is uniformly bounded by $M$, $rank(\widehat{\nabla_{\boldsymbol{\theta}}\psi}) = rank(\nabla_{\boldsymbol{\theta}}\psi)$ and $\left\|\widehat{\nabla_{\boldsymbol{\theta}}\psi} - \nabla_{\boldsymbol{\theta}}\psi\right\|_2 \cdot \kappa_{\nabla_{\boldsymbol{\theta}}\psi} < \|\nabla_{\boldsymbol{\theta}}\psi\|_2$. Then with probability $1 - \delta$:*

$$\left\|\boldsymbol{\omega}^L - \hat{\boldsymbol{\omega}}\right\|_2 \leq M\sqrt{qd}\sqrt{\frac{\log(\frac{2}{\delta})}{2n}}\left(\frac{\kappa_{\nabla_{\boldsymbol{\theta}}\psi}\left\|\boldsymbol{\omega}^L\right\|_2}{c\left\|\nabla_{\boldsymbol{\theta}}\psi\right\|_2} + \|y\|_2\right), \tag{12}$$

*where $\boldsymbol{\omega}^L$ are the real reward parameters and $\hat{\omega}$ are the parameters recovered with Equation (5), $c = 1 - \frac{\left\|\widehat{\nabla_{\boldsymbol{\theta}}\psi} - \nabla_{\boldsymbol{\theta}}\psi\right\|_2}{\|\nabla_{\boldsymbol{\theta}}\psi\|_2}\kappa_{\nabla_{\boldsymbol{\theta}}\psi} > 0$, and $y = \nabla_{\boldsymbol{\theta}}\psi^{+H}\boldsymbol{\omega}$.*

*Proof.* We need to bound the difference in norm between $\boldsymbol{\omega}^L$ and $\hat{\boldsymbol{\omega}}$ that are the true parameters and the parameters that we recovered solving the minimization problem (4).

$$\left\|\boldsymbol{\omega}^L - \widehat{\boldsymbol{\omega}}\right\|_2 \quad (13)$$

$$\leq \frac{\kappa}{\left(1 - \kappa \frac{\|\delta\nabla_{\boldsymbol{\theta}}\boldsymbol{\psi}\|_2}{\|\nabla_{\boldsymbol{\theta}}\boldsymbol{\psi}\|_2}\right)\|\nabla_{\boldsymbol{\theta}}\boldsymbol{\psi}\|_2} \left( \frac{\|\delta\nabla_{\boldsymbol{\theta}}\boldsymbol{\psi}\|_2}{\|\nabla_{\boldsymbol{\theta}}\boldsymbol{\psi}\|_2} \left\|\boldsymbol{\omega}^L\right\|_2 \|\nabla_{\boldsymbol{\theta}}\boldsymbol{\psi}\|_2 \right) + \frac{\|\delta\nabla_{\boldsymbol{\theta}}\boldsymbol{\psi}\|_2}{\|\nabla_{\boldsymbol{\theta}}\boldsymbol{\psi}\|_2} \|y\|_2 \|\nabla_{\boldsymbol{\theta}}\boldsymbol{\psi}\|_2 \quad (14)$$

$$\leq \frac{\kappa}{c\,\|\nabla_{\boldsymbol{\theta}}\boldsymbol{\psi}\|_2} \left( \frac{\|\delta\nabla_{\boldsymbol{\theta}}\boldsymbol{\psi}\|_2}{\|\nabla_{\boldsymbol{\theta}}\boldsymbol{\psi}\|_2} \left\|\boldsymbol{\omega}^L\right\|_2 \|\nabla_{\boldsymbol{\theta}}\boldsymbol{\psi}\|_2 \right) + \frac{\|\delta\nabla_{\boldsymbol{\theta}}\boldsymbol{\psi}\|_2}{\|\nabla_{\boldsymbol{\theta}}\boldsymbol{\psi}\|_2} \|y\|_2 \|\nabla_{\boldsymbol{\theta}}\boldsymbol{\psi}\|_2 \quad (15)$$

$$= \|\delta\nabla_{\boldsymbol{\theta}}\boldsymbol{\psi}\|_2 \left( \frac{\kappa\left\|\boldsymbol{\omega}^L\right\|_2}{c\,\|\nabla_{\boldsymbol{\theta}}\boldsymbol{\psi}\|_2} + \|y\|_2 \right) \quad (16)$$

$$\leq M\sqrt{qd}\sqrt{\frac{\log(\frac{2}{\delta})}{2n}} \left( \frac{\kappa\left\|\boldsymbol{\omega}^L\right\|_2}{c\,\|\nabla_{\boldsymbol{\theta}}\boldsymbol{\psi}\|_2} + \|y\|_2 \right), \quad (17)$$

where line (14) is obtained by using Lemma A.1, lines (15, 16) by rearranging the terms, and line (17) by using Lemma A.4. We can observe that the last term vanishes when the rank($\nabla_{\boldsymbol{\theta}}\boldsymbol{\psi}$)= $q$ (see [39]). $\qquad\square$

### A.3 Proofs and derivation of Section 4

Now we will give the proofs and derivations of Lemmas 4.1 and Theorem 4.1.

**Lemma 4.1.** *If the matrix $\left(\sum_{t=1}^{m}\alpha_t^2\nabla_{\boldsymbol{\theta}}\boldsymbol{\psi}_t^T\nabla_{\boldsymbol{\theta}}\boldsymbol{\psi}_t\right)^{-1}$ is full-rank than optimization problem (4) is solved in closed form by*

$$\widehat{\boldsymbol{\omega}} = \left(\sum_{t=1}^{m}\alpha_t^2\nabla_{\boldsymbol{\theta}}\boldsymbol{\psi}_t^T\nabla_{\boldsymbol{\theta}}\boldsymbol{\psi}_t\right)^{-1}\left(\sum_{t=1}^{m}\alpha_t\nabla_{\boldsymbol{\theta}}\boldsymbol{\psi}_t^T\Delta_t\right). \quad (5)$$

*Proof.* Taking the derivative of (4) with respect to $\omega$:

$$\nabla_{\boldsymbol{\omega}}\sum_{t=1}^{m}\|\Delta_t - \alpha_t\nabla_{\boldsymbol{\theta}}\boldsymbol{\psi}_t\boldsymbol{\omega}\|_2^2 = \sum_{t=1}^{m}\nabla_{\boldsymbol{\omega}}(\Delta_t - \alpha_t\nabla_{\boldsymbol{\theta}}\boldsymbol{\psi}_t\boldsymbol{\omega})^T(\Delta - \alpha\nabla_{\boldsymbol{\theta}}\boldsymbol{\psi}_t\boldsymbol{\omega})$$

$$= \sum_{t=1}^{m}\nabla_{\boldsymbol{\omega}}(\Delta_t^T\Delta_t + (\alpha\nabla_{\boldsymbol{\theta}}\boldsymbol{\psi}_t\boldsymbol{\omega})^T(\alpha_t\nabla_{\boldsymbol{\theta}}\boldsymbol{\psi}_t\boldsymbol{\omega}) - 2\alpha_t\nabla_{\boldsymbol{\theta}}\boldsymbol{\psi}_t\boldsymbol{\omega})^T\Delta_t)$$

$$= 2\left(\sum_{t=1}^{m}\alpha_t^2\nabla_{\boldsymbol{\theta}}\boldsymbol{\psi}_t^T\nabla_{\boldsymbol{\theta}}\boldsymbol{\psi}_t\right)\boldsymbol{\omega} - 2\sum_{t=1}^{m}\left(\alpha_t\nabla_{\boldsymbol{\theta}}\boldsymbol{\psi}_t^T\Delta_t\right).$$

Taking it equal to zero:

$$\left(\sum_{t=1}^{m}\alpha_t^2\nabla_{\boldsymbol{\theta}}\boldsymbol{\psi}_t^T\nabla_{\boldsymbol{\theta}}\boldsymbol{\psi}_t\right)\boldsymbol{\omega} - \sum_{t=1}^{m}\left(\alpha_t\nabla_{\boldsymbol{\theta}}\boldsymbol{\psi}_t^T\Delta_t\right) = 0$$

$$\boldsymbol{\omega} = \left(\sum_{t=1}^{m}\alpha_t^2\nabla_{\boldsymbol{\theta}}\boldsymbol{\psi}_t^T\nabla_{\boldsymbol{\theta}}\boldsymbol{\psi}_t\right)^{-1}\left(\sum_{t=1}^{m}\alpha_t\nabla_{\boldsymbol{\theta}}\boldsymbol{\psi}_t^T\Delta_t\right)$$

$\qquad\square$

**Lemma A.5.** *The regularized version of (4) is equal to:*

$$\min_{\boldsymbol{\omega}}\sum_{t=1}^{m}\|\Delta_t - \alpha_t\nabla_{\boldsymbol{\theta}}\boldsymbol{\psi}_t\boldsymbol{\omega}\|_2^2 + \lambda\,\|\boldsymbol{\omega}\|_2^2,$$

*where $\lambda > 0$. We can solve the regularized problem in closed form:*

$$\boldsymbol{\omega} = \left(\sum_{t=1}^{m}\alpha_t^2\nabla_{\boldsymbol{\theta}}\boldsymbol{\psi}_t^T\nabla_{\boldsymbol{\theta}}\boldsymbol{\psi}_t + \lambda\mathbf{I}_d\right)^{-1}\left(\sum_{t=1}^{m}\alpha_t\nabla_{\boldsymbol{\theta}}\boldsymbol{\psi}_t^T\Delta_t\right).$$

*Proof.* Taking the derivative respect to $\omega$:

$$\nabla_{\boldsymbol{\omega}} \sum_{t=1}^{m} \|\Delta_t - \alpha_t \nabla_{\boldsymbol{\theta}} \psi_t \boldsymbol{\omega}\|_2^2 + \lambda \|\boldsymbol{\omega}\|_2^2 = \sum_{t=1}^{m} \nabla_{\boldsymbol{\omega}} (\Delta_t - \alpha_t \nabla_{\boldsymbol{\theta}} \psi_t \boldsymbol{\omega})^T (\Delta_t - \alpha \nabla_{\boldsymbol{\theta}} \psi_t \boldsymbol{\omega}) + \nabla_{\boldsymbol{\omega}} \lambda \boldsymbol{\omega}^T \boldsymbol{\omega}$$

$$= \sum_{t=1}^{m} \nabla_{\boldsymbol{\omega}} (\Delta_t^T \Delta_t + (\alpha \nabla_{\boldsymbol{\theta}} \psi_t \boldsymbol{\omega})^T (\alpha_t \nabla_{\boldsymbol{\theta}} \psi_t \boldsymbol{\omega}) - 2\alpha_t \nabla_{\boldsymbol{\theta}} \psi_t \boldsymbol{\omega})^T \Delta_t) + 2\lambda \boldsymbol{\omega}$$

$$= 2 \left( \sum_{t=1}^{m} \alpha_t^2 \nabla_{\boldsymbol{\theta}} \psi_t^T \nabla_{\boldsymbol{\theta}} \psi_t \right) \boldsymbol{\omega} - 2 \sum_{t=1}^{m} \left( \alpha_t \nabla_{\boldsymbol{\theta}} \psi_t^T \Delta_t \right) + 2\lambda \boldsymbol{\omega}.$$

Taking it equal to zero:

$$\left( \sum_{t=1}^{m} \alpha_t^2 \nabla_{\boldsymbol{\theta}} \psi_t^T \nabla_{\boldsymbol{\theta}} \psi_t + \lambda \mathbf{I}_d \right) \boldsymbol{\omega} - \sum_{t=1}^{m} \left( \alpha_t \nabla_{\boldsymbol{\theta}} \psi_t^T \Delta_t \right) = 0$$

$$\boldsymbol{\omega} = \left( \sum_{t=1}^{m} \alpha_t^2 \nabla_{\boldsymbol{\theta}} \psi_t^T \nabla_{\boldsymbol{\theta}} \psi_t + \lambda \mathbf{I}_d \right)^{-1} \left( \sum_{t=1}^{m} \alpha_t \nabla_{\boldsymbol{\theta}} \psi_t^T \Delta_t \right).$$

$\square$

**Theorem 4.1.** *Let* $\boldsymbol{\Psi}$ *be the real Jacobians and* $\widehat{\boldsymbol{\Psi}}$ *the estimated Jacocobian from $n$ trajectories* $\{\tau_1, \cdots, \tau_n\}$. *Assume that* $\boldsymbol{\Psi}$ *is bounded by a constant $M$ and* $\lambda_{\min}(\widehat{\boldsymbol{\Psi}}^T \widehat{\boldsymbol{\Psi}}) \geq \lambda > 0$. *Then w.h.p.:*

$$\left\| \boldsymbol{\omega}^L - \widehat{\boldsymbol{\omega}} \right\|_2 \leq O \left( \frac{1}{\lambda} M \sqrt{\frac{dq \log(\frac{2}{\delta})}{2n}} \left( \sqrt{\frac{\log dq}{m}} + \sqrt{dq} \right) \right).$$

*Proof.* We decompose the estimated Jacobian $\boldsymbol{\Psi} = \boldsymbol{\Psi} + E$, where $E$ is the random variable component caused by the estimation of the $\nabla_{\boldsymbol{\theta}} \psi$. Since we estimate the jacobians with an unbiased estimator the mean of $E$ is 0. We reshape the $\boldsymbol{\Psi}$ and $E$ as $\boldsymbol{\Psi} \in \mathbb{R}^{m \times dq}$ and $E \in \mathbb{R}^{m \times dq}$. Now $E$, since its mean is 0 and all lines are independent of each other, is a sub-Gaussian matrix with parameters $(\frac{1}{m} \Sigma_E, \frac{1}{m} \sigma_E)$. The proof is similar to the proof of Theorem 1 in [18].

$$\left\| (\boldsymbol{\Psi} + E)^T (\boldsymbol{\Psi}^T \boldsymbol{\omega}^L) - (\boldsymbol{\Psi} + E)^T (\boldsymbol{\Psi} + E) \boldsymbol{\omega}^L \right\|_2$$

$$= \left\| \boldsymbol{\Psi}^T \nabla_{\boldsymbol{\theta}} \psi^T \boldsymbol{\omega}^* + E^T \boldsymbol{\Psi}^T \boldsymbol{\omega}^L - \boldsymbol{\Psi}^T \boldsymbol{\Psi} \boldsymbol{\omega}^L - \boldsymbol{\Psi}^T E \boldsymbol{\omega}^* - E \boldsymbol{\Psi}^T \boldsymbol{\omega}^* - E^T E \boldsymbol{\omega}^L \right\|_2$$

$$= \left\| -\boldsymbol{\Psi}^T E \boldsymbol{\omega}^L - E^T E \boldsymbol{\omega}^L \right\|_2.$$

Now we bound separately these two terms, using Lemma A.3 as in [18]:

$$\left\| \boldsymbol{\Psi}^T E \boldsymbol{\omega} \right\|_2 \leq \|\boldsymbol{\Psi}\|_2 \sigma_E \|\boldsymbol{\omega}^L\|_2 \sqrt{\frac{\log dq}{m}}$$

$$\left\| E^T E \boldsymbol{\omega}^L \right\|_2 = \left\| (E^T E + \sigma_E^2 \mathbf{I}_{qd} - \sigma_E^2 \mathbf{I}_{qd}) \boldsymbol{\omega}^L \right\|_2 \leq \sigma_E^2 \left( C \sqrt{\frac{\log dq}{m}} + \sqrt{dq} \right) \|\boldsymbol{\omega}^L\|_2$$

with probability $1 - c_1 \exp(-c_2 \log q)$ where $c_1, c_2$ are positive constants that do not depend on $\sigma_E, n, q$. So now applying Lemma A.2:

$$\left\| \boldsymbol{\omega}^L - \widehat{\boldsymbol{\omega}} \right\|_2 \leq \frac{1}{\lambda} \left( \|\boldsymbol{\Psi}\|_2 \sigma_E \|\boldsymbol{\omega}^L\|_2 \sqrt{\frac{\log dq}{m}} + \sigma_E^2 \left( C \sqrt{\frac{\log dq}{m}} + \sqrt{dq} \right) \|\boldsymbol{\omega}^L\|_2 \right)$$

w.h.p..

Now we need to bound the random variable $\sigma_E$. Remember that $E_i = \nabla_{\boldsymbol{\theta}} \psi_i - \widehat{\nabla_{\boldsymbol{\theta}} \psi}_i$. Since $\widehat{\nabla_{\boldsymbol{\theta}} \psi}$ are assumed to be bounded by $M$, by applying Hoeffding's inequality, with probability $1 - \delta_1$:

$$\|E_i\|_2 = \left\| \widehat{\nabla_{\boldsymbol{\theta}} \psi}_i - \nabla_{\boldsymbol{\theta}} \psi_i \right\|_2 \leq M \sqrt{\frac{dq \log(\frac{2}{\delta_1})}{2n}}.$$

So $E$ is a subgaussian random variable where each component is bounded by $M\sqrt{\frac{dq\log(\frac{2}{\delta_1})}{2n}}$.

Then:

$$\mathrm{P}\left[\left\|\boldsymbol{\omega}^L - \widehat{\boldsymbol{\omega}}\right\|_2 \geq \frac{1}{\lambda}M\sqrt{\frac{dq\log(\frac{2}{\delta_1})}{2n}}\left\|\boldsymbol{\omega}^L\right\|_2\|\boldsymbol{\Psi}\|_2\sqrt{\frac{\log dq}{m}} + M\frac{dq\log(\frac{2}{\delta_1})}{2n}C\sqrt{\frac{\log dq}{m}} + \sqrt{dq}\right]$$

$$\leq \mathrm{P}\left[\left\|\boldsymbol{\omega}^L - \widehat{\boldsymbol{\omega}}\right\|_2 \geq \frac{1}{\lambda}M\sqrt{\frac{dq\log(\frac{2}{\delta_1})}{2n}}\left\|\boldsymbol{\omega}^L\right\|_2\left(\|\boldsymbol{\Psi}\|_2\sqrt{\frac{\log dq}{m}} + C\sqrt{\frac{\log dq}{m}} + \sqrt{dq}\right)\right]$$

$$\leq \delta_1 + c1\exp(-c2\log dq)$$

So the result follows where with w.h.p. we mean with probability $1 - (\delta_1 + c1\exp(-c2\log dq))$ as in [18]. $\qquad\square$

### A.4  Proofs of Section 5

In this section we provide the proofs and derivations of the theorems in Section 5.

**Lemma A.6.** *Given a dataset $D = \{(s_1, a_1), \cdots, (s_n, a_m)\}$ of state-action couples sampled from a Gaussian linear policy $\pi_{\boldsymbol{\theta}}(\cdot|s) \sim \mathcal{N}(\boldsymbol{\theta}^T\boldsymbol{\varphi}(s), \sigma^2)$ such that $S \in \mathbb{R}^{n\times p}$ is the matrix of states features and let the minimum singular value of $(S^T S)$ $\sigma_{\min} \geq \eta$, then the error between the maximum likelihood estimator $\boldsymbol{\theta}^{MLE}$ and the mean $\boldsymbol{\theta}$ is, with probability $1 - \delta$:*

$$\left\|\boldsymbol{\theta}^{MLE} - \boldsymbol{\theta}\right\|_2 \leq \sigma\sqrt{\frac{r + \log(\frac{1}{\delta})}{n\eta}},$$

*where $r$ is the rank($S^T S$).*

*Proof.* We start by stating that the maximum likelihood for linear Gaussian policies can be recast as an ordinary least-squares problem. We write the Likelihood $L(\boldsymbol{\theta})$

$$\log L(\boldsymbol{\theta}) = \log\left(\prod_{i=1}^n \pi(a_i|s_i)\right)$$

$$= \sum_{i=1}^n \log\left(\frac{1}{\sqrt{2\pi\sigma^2}}\exp\left(-\frac{(a_i - \boldsymbol{\theta}^T\boldsymbol{\varphi}(s_i))^2}{2\sigma^2}\right)\right)$$

$$= n\log\left(\frac{1}{\sqrt{2\pi\sigma^2}}\right) - \sum_{i=1}^n \frac{(a_i - \boldsymbol{\theta}^T\boldsymbol{\varphi}(s_i))^2}{2\sigma^2}$$

The resulting maximum likelihood problem is given by:

$$\max_{\boldsymbol{\theta}} \log L(\boldsymbol{\theta}) = \min_{\boldsymbol{\theta}} \sum_{i=1}^n (a_i - \boldsymbol{\theta}^T\boldsymbol{\varphi}(s_i))^2$$

So we have the following linear least-squares problem:

$$\min_{\theta} \|S\boldsymbol{\theta} - A + \epsilon\|_2,$$

where $\epsilon$ is an error with mean $0$ and variance $\sigma^2$, $S \in \mathbb{R}^{n\times d}$ is the matrix of states features and $A \in \mathbb{R}^n$ is the vector of actions. Using Theorem A.1, we can say that with probability $1 - \delta$:

$$\left\|\boldsymbol{\theta}^{\mathrm{MLE}} - \boldsymbol{\theta}\right\|_2 \leq \sigma\sqrt{\frac{r + \log(\frac{1}{\delta})}{n\eta}},$$

where $r$ is the rank($S^T S$). $\qquad\square$

**Lemma A.7.** *Given two Gaussian policies $\pi_{\boldsymbol{\theta}_1}(\cdot|s) \sim \mathcal{N}(\boldsymbol{\theta}_1^T\boldsymbol{\varphi}(s), \sigma^2)$ and $\pi_{\boldsymbol{\theta}_2}(\cdot|s) \sim \mathcal{N}(\boldsymbol{\theta}_2^T\boldsymbol{\varphi}(s), \sigma^2)$ with same variance and the state features are bounded by $M_S$:*

$$\left\|\nabla_{\boldsymbol{\theta}}\log\pi_{\boldsymbol{\theta}_1}(a|s) - \nabla_{\boldsymbol{\theta}}\log\pi_{\boldsymbol{\theta}_2}(a|s)\right\|_2 \leq \frac{M_S^2}{\sigma^2}\left\|\boldsymbol{\theta}_1 - \boldsymbol{\theta}_2\right\|_2.$$

*Proof.* The gradient of the log policy of a general policy $\pi_{\boldsymbol{\theta}}(a|s)$ is:

$$\nabla_{\boldsymbol{\theta}} \log \pi(a|s) = \frac{\boldsymbol{\varphi}(s)^T (a - \boldsymbol{\theta}^T \boldsymbol{\varphi}(s))}{\sigma^2}.$$

Now we apply this result to the difference in norm between two Gaussian log policies:

$$\left\| \nabla_{\boldsymbol{\theta}} \log \pi_{\boldsymbol{\theta}_1}(a|s) - \nabla_{\boldsymbol{\theta}} \log \pi_{(a|s)} \right\|_2 = \left\| \frac{\boldsymbol{\varphi}(s)^T (a - \boldsymbol{\theta}_1^T \boldsymbol{\varphi}(s))}{\sigma^2} - \frac{\boldsymbol{\varphi}(s)^T (a - \boldsymbol{\theta}_2^T \boldsymbol{\varphi}(s))}{\sigma^2} \right\|_2 \quad (18)$$

$$= \left\| \frac{\boldsymbol{\varphi}(s)}{\sigma^2} (\boldsymbol{\theta}_1^T \boldsymbol{\varphi}(s) - \boldsymbol{\theta}_2^T \boldsymbol{\varphi}(s)) \right\|_2 \quad (19)$$

$$\leq \left\| \frac{\boldsymbol{\varphi}(s)}{\sigma^2} \right\|_2 \left\| \boldsymbol{\theta}_1 - \boldsymbol{\theta}_2 \right\|_2 \left\| \boldsymbol{\varphi}(s) \right\|_2 \quad (20)$$

$$\leq \frac{M_S^2}{\sigma^2} \left\| \boldsymbol{\theta}_1 - \boldsymbol{\theta}_2 \right\|_2 . \quad (21)$$

In line (19) we use the Cauchy-Schwartz inequality, and in line (20) the assumption that the state features are bounded by $M_S$. $\qquad\square$

**Lemma A.8.** *Given a dataset $D = \{\tau_1, \cdots, \tau_n\}$ of trajectories such that every trajectory $\tau_i = \{(s_1, a_1), \cdots, (s_T, a_T)\}$ is sampled from a Gaussian linear policy $\pi_{\boldsymbol{\theta}}(\cdot|s) \sim \mathcal{N}(\boldsymbol{\theta}^T \boldsymbol{\varphi}(s), \sigma)$, the maximum likelihood estimator $\boldsymbol{\theta}^{MLE}$ estimated on $D$, the condition of Lemma A.6 holds, the $\widehat{\nabla_{\boldsymbol{\theta}} \psi}$ uniformly bounded by $M$, the state features bounded by $M_S$, the reward features bounded by $M_R$. Let $S \in \mathbb{R}^{n \times p}$ be the matrix of state features and let $\sigma_{\min}(S^T S) \geq \eta$. Then with probability $1 - \delta$:*

$$\left\| \widehat{\nabla_{\boldsymbol{\theta}} \psi}(\boldsymbol{\theta}^{MLE}) - \nabla_{\boldsymbol{\theta}} \psi(\boldsymbol{\theta}) \right\|_2 \leq M \sqrt{qd} \sqrt{\frac{\log(\frac{2}{\delta})}{2n}} + \frac{T M_S^2 M_R}{(1-\gamma)\sigma} \sqrt{\frac{r + \log(\frac{1}{\delta})}{n\eta}},$$

*where $\gamma$ is the discount factor and $r$ is the rank of $S^T S$.*

*Proof.* We start by decomposing the norm of the difference in two components, using triangular inequality:

$$\left\| \widehat{\nabla_{\boldsymbol{\theta}} \psi}(\widehat{\boldsymbol{\theta}}) - \nabla_{\boldsymbol{\theta}} \psi(\boldsymbol{\theta}) \right\|_2 \leq \left\| \widehat{\nabla_{\boldsymbol{\theta}} \psi}(\boldsymbol{\theta}) - \nabla_{\boldsymbol{\theta}} \psi(\boldsymbol{\theta}) \right\|_2 + \left\| \widehat{\nabla_{\boldsymbol{\theta}} \psi}(\boldsymbol{\theta}) - \widehat{\nabla_{\boldsymbol{\theta}} \psi}(\widehat{\boldsymbol{\theta}}) \right\|_2 .$$

The first component is bounded by Lemma A.4. We will bound now the second component, using Reinforce estimator for the gradient:

$$\left\| \widehat{\nabla_{\boldsymbol{\theta}} \psi}(\boldsymbol{\theta}) - \widehat{\nabla_{\boldsymbol{\theta}} \psi}(\widehat{\boldsymbol{\theta}}) \right\|_2 = \quad (22)$$

$$= \left\| \frac{1}{n} \sum_{i=1}^n \sum_{t=1}^T \nabla_{\boldsymbol{\theta}} \log \pi_{\boldsymbol{\theta}}(a_{i,t}|s_{i,t}) R_{i,t} \gamma^t - \frac{1}{n} \sum_{i=1}^n \sum_{t=1}^T \nabla_{\boldsymbol{\theta}} \log \pi_{\widehat{\boldsymbol{\theta}}}(a_{i,t}|s_{i,t}) R_{i,t} \gamma^t \right\|_2 \quad (23)$$

$$= \frac{1}{n} \left\| \sum_{i=1}^n \sum_{t=1}^T (\nabla_{\boldsymbol{\theta}} \log \pi_{\boldsymbol{\theta}}(a_{i,t}|s_{i,t}) - \nabla_{\boldsymbol{\theta}} \log \pi_{\widehat{\boldsymbol{\theta}}}(a_{i,t}|s_{i,t})) R_{i,t} \gamma^t \right\|_2 \quad (24)$$

$$\leq \frac{1}{n} \sum_{i=1}^n \sum_{t=1}^T \left\| (\nabla_{\boldsymbol{\theta}} \log \pi_{\boldsymbol{\theta}}(a_{i,t}|s_{i,t}) - \nabla_{\boldsymbol{\theta}} \log \pi_{\widehat{\boldsymbol{\theta}}}(a_{i,t}|s_{i,t})) \right\|_2 \left\| R_t \gamma^t \right\|_2 \quad (25)$$

$$\leq \frac{1}{n} \frac{M_R}{(1-\gamma)} \sum_{i=1}^n \sum_{t=1}^T \frac{M_S^2}{\sigma^2} \left\| \boldsymbol{\theta} - \widehat{\boldsymbol{\theta}} \right\|_2 \quad (26)$$

$$\leq \frac{T M_S^2 M_R}{\sigma^2 (1-\gamma)} \sigma \sqrt{\frac{r + \log(\frac{1}{\delta})}{n\eta}}. \quad (27)$$

In line (26) we apply the Cauchy-Schwartz inequality. In line (27) we apply lemma A.4 and in line (28) we apply lemma A.4. Merging the two results the proof follows. $\qquad\square$

**Theorem 5.1.** *Let $\pi_{\boldsymbol{\theta}_1}, \pi_{\boldsymbol{\theta}_2}$ be two Gaussian policies $\pi_{\boldsymbol{\theta}_i}(\cdot|s) \sim \mathcal{N}(\boldsymbol{\theta}_i^T \boldsymbol{\varphi}(s), \sigma^2)$ with $i \in \{1, 2\}$, such that $\pi_{\boldsymbol{\theta}_2}$ is the improvement of $\pi_{\boldsymbol{\theta}_1}$. Let $i \in [1, 2]$. Given datasets $D_i = \{\tau_1^i, \ldots, \tau_n^i\}$ of trajectories generated by $\pi_i$, such that $S_i \in \mathbb{R}^{n \times t \times d}$ is the matrix of corresponding states features, let the minimum singular value of $\sigma_{\min}(S_i^T S_i) \geq \eta > 0$, $\widehat{\nabla_{\boldsymbol{\theta}_1} \boldsymbol{\psi}}$ uniformly bounded by $M$, the state features bounded by $M_S$, and the reward features bounded by $M_R$. Then with probability $1 - \delta$:*

$$\left\| \boldsymbol{\omega}^L - \widehat{\boldsymbol{\omega}} \right\|_2 \leq O\left( \frac{(M + M_S^2 M_R)}{\sigma_{\min}(\nabla_{\boldsymbol{\theta}_1} \boldsymbol{\psi})} \sqrt{\frac{\log(\frac{2}{\delta})}{n\eta}} \right)$$

*where $\boldsymbol{\omega}^L$ are the real reward parameters and $\widehat{\omega}$ are the parameters recovered using Lemma 4.1.*

*Proof.* First we have to bound the error on $\Delta$ created by the behavioral cloning. Given $\Delta = \boldsymbol{\theta}_2 - \boldsymbol{\theta}_1$ and $\widehat{\Delta} = \widehat{\boldsymbol{\theta}}_2 - \widehat{\boldsymbol{\theta}}_1$:

$$\left\| \Delta - \widehat{\Delta} \right\| = \left\| \boldsymbol{\theta}_2 - \boldsymbol{\theta}_1 - \widehat{\boldsymbol{\theta}}_2 + \widehat{\boldsymbol{\theta}}_1 \right\| \leq \left\| \boldsymbol{\theta}_1 - \widehat{\boldsymbol{\theta}}_1 \right\| + \left\| \boldsymbol{\theta}_2 - \widehat{\boldsymbol{\theta}}_2 \right\| \leq 2\sigma \sqrt{\frac{r + \log(\frac{1}{\delta})}{n\eta}}. \tag{28}$$

So we can bound the difference in norm between the real weights $\boldsymbol{\omega}^L$ and the estimated weights $\widehat{\omega}$. We indicate with $\kappa$ the condition number of $\nabla_{\boldsymbol{\theta}} \boldsymbol{\psi}$, with $\chi = \frac{\left\| \widehat{\nabla_{\boldsymbol{\theta}} \boldsymbol{\psi}} - \nabla_{\boldsymbol{\theta}} \boldsymbol{\psi} \right\|_2}{\|\nabla_{\boldsymbol{\theta}} \boldsymbol{\psi}\|_2}$ and $y = \nabla_{\boldsymbol{\theta}} \boldsymbol{\psi}^{+H} \boldsymbol{\omega}$. We apply the perturbation Lemma A.1.

$$\left\| \boldsymbol{\omega}^L - \widehat{\boldsymbol{\omega}} \right\|_2 \leq \frac{\kappa}{(1 - \kappa\chi) \|\nabla_{\boldsymbol{\theta}} \boldsymbol{\psi}\|_2} \left( \chi \left\| \boldsymbol{\omega}^L \right\|_2 \|\nabla_{\boldsymbol{\theta}} \boldsymbol{\psi}\|_2 + \left\| \Delta - \widehat{\Delta} \right\|_2 \right) + \chi \|y\|_2 \|\nabla_{\boldsymbol{\theta}} \boldsymbol{\psi}\|_2 \tag{29}$$

$$\leq \frac{\kappa \left\| \widehat{\nabla_{\boldsymbol{\theta}} \boldsymbol{\psi}} - \nabla_{\boldsymbol{\theta}} \boldsymbol{\psi} \right\|_2}{c \|\nabla_{\boldsymbol{\theta}} \boldsymbol{\psi}\|_2} \left\| \boldsymbol{\omega}^L \right\|_2 + \frac{\kappa \left\| \Delta - \widehat{\Delta} \right\|_2}{c \|\nabla_{\boldsymbol{\theta}} \boldsymbol{\psi}\|_2} + \|\nabla_{\boldsymbol{\theta}} \boldsymbol{\psi}\|_2 \|y\|_2$$

$$= \left\| \nabla_{\boldsymbol{\theta}} \boldsymbol{\psi} - \widehat{\nabla_{\boldsymbol{\theta}} \boldsymbol{\psi}} \right\|_2 \left( \frac{\kappa \left\| \boldsymbol{\omega}^L \right\|_2}{c \|\nabla_{\boldsymbol{\theta}} \boldsymbol{\psi}\|_2} + \|y\|_2 \right) + \left\| \Delta - \widehat{\Delta} \right\|_2 \frac{\kappa}{c \|\nabla_{\boldsymbol{\theta}} \boldsymbol{\psi}\|_2}$$

$$= \left\| \nabla_{\boldsymbol{\theta}} \boldsymbol{\psi} - \widehat{\nabla_{\boldsymbol{\theta}} \boldsymbol{\psi}} \right\|_2 \left( \frac{\left\| \boldsymbol{\omega}^L \right\|_2}{c \sigma_{\min}(\nabla_{\boldsymbol{\theta}} \boldsymbol{\psi})} + \|y\|_2 \right) + \left\| \Delta - \widehat{\Delta} \right\|_2 \frac{1}{c \sigma_{\min}(\nabla_{\boldsymbol{\theta}} \boldsymbol{\psi})}$$

$$+ 2\sigma \sqrt{\frac{r + \log(\frac{1}{\delta})}{n\eta}} \frac{1}{c \sigma_{\min}(\nabla_{\boldsymbol{\theta}} \boldsymbol{\psi})} \tag{30}$$

$$\leq O\left( \frac{(M + M_S^2 M_R)}{\sigma_{\min}(\nabla_{\boldsymbol{\theta}} \boldsymbol{\psi})} \sqrt{\frac{\log(\frac{2}{\delta})}{n\eta}} \right), \tag{31}$$

where in line (29) we apply Lemma A.1 and in line (30) we apply Equation (28) and Lemma A.8. □

# B Experiments

In this appendix, we report some experimental details together with some additional experiments.

## B.1 Gridworld

In the Gridworld experiment, we select different learning steps for different learners. The number of learning steps depends on the number of policy updates that the learner takes to become an expert. In the following plots, we report the expected discounted return for each learner: Q-Learning (Figure 6), G(PO)MDP (Figure 5), SPI (Figure 7), SVI (Figure 8). In these plots, the expected discounted return is estimating using a batch of 50 trajectories for each learner. The discount factor used in all experiments is 0.96.

Figure 5: Learning performance of G(PO)MDP. 20 runs, 98%c.i.

Figure 6: Learning performance of Q-Learning. 20 runs, 98%c.i.

Figure 7: Learning performance of SPI. 20 runs, 98%c.i.

Figure 8: Learning performance of SVI. 20 runs, 98%c.i.

## B.2 MuJoCo additional experiments

For the MuJoCo experiments, we use the same hyperparameters as in [17], apart from that we use 16 parallel agents for PPO, due to resource constraints. The number of forward steps are settled to 2000. As in [17], we select a subset of the learner's trajectories and we do not use the first 10 trajectories because the first phase of learning is too noisy. We evaluate the algorithms on the first 1 million environment steps.

|       | Recovered Weights | Real Weights |
|-------|-------------------|--------------|
| Time  | 0.0401            | 0.0017       |
| Jerk  | 0.0174            | 0.0003       |
| Slow  | 0.0001            | 0.0000       |
| Crash | 0.9424            | 0.9980       |

Table 1: Reward weights for the autonomous simulate driving scenario.

Figure 9: Reward weights for the autonomous simulate driving scenario.

## B.3 Autonomous driving scenario

In this section, we report an additional, preliminary experiment, that we perform on a simulator driving scenario. We employ SUMO simulator, an open-source, continuous road traffic simulation package designed to handle large road networks. SUMO focuses on the high-level control of the car, integrating an internal system that controls the vehicle dynamics. During the simulation, SUMO provides information on the other vehicles around the ego vehicle.

We consider a crossroad scenario which consists of an intersection with an arbitrary number of roads. The vehicle coming from the source road has to reach a target road that has a higher priority. The goal of the agent is to drive the ego car and enter the target road, avoiding dangerous manoeuvres.

The reward features consists of four components: *Time*, a constant feature at each decision step; *Jerk*, the absolute value of the instantaneous jerk, i.e., the finite- difference derivative of the acceleration; *Harsh Slow Down*, a binary feature, which activates whenever the velocity is lower than a threshold; *Crash*, a binary feature which activates when the vehicle violates the safety constraints or performs a crash.

The agent's policy is a rule-based policy, i.e., a set of parametrized rules, which is learned using Policy Gradients with Parameter-based Exploration (PGPE). It is important to notice that the agent's policy is not differentiable.

We perform 10 PGPE updates of the agents and then we use the learning trajectories with LOGEL. In the behavioural cloning step, we use a linear layer to approximate the policy of the learner. Table 1 shows the normalized weights recovered by LOGEL and the normalized real weights. As shown in Table 1 and in Figure 9, the reward weights recovered are quite similar to the real ones.

We thank Amarildo Likmeta to provide us the data to perform this experiment.