[Reviews · NeurIPS 2020]

Review 1

Summary and Contributions: AUTHOR RESPONSE: Thanks for the detailed author response! My original concerns were (1) the usefulness of the proposed problem setting and (2) whether the proposed method actually recovers the expert's behavior. I appreciate the pointer to the autonomous-driving experiment in the Appendix. It's not quite what I was looking for: I was looking for evidence that the proposed method works on data generated from a human (or a high-fidelity simulator of human learning). In retrospect, I realize that my question was unclear, so I won't count this against the paper. The additional experiment in Fig 3 addresses concern (2). I therefore increase my review (5 -> 6). The motivation for not increasing the score further is R4's concern that just doing IRL or BC on the learner's experience after learning might be a very strong baseline. -------------------------------------------- This paper studies inverse RL in the setting where you observe an agent's behavior over the course of learning. The proposed method assumes that the agent is being updated using a gradient-based RL algorithm, and uses the observed behavior to infer the policy parameters at each stage of learning (via behavior cloning), and then infers the reward function by "inverting" the policy gradient. The paper presents finite-sample bounds for the method, and shows results on both gridworlds some two continuous control tasks.

Strengths: The idea is explained clearly, and the finite-sample bounds strengthen the paper. The problem setting is novel (though it remains to be seen how useful it is). To the best of my knowledge, the idea of inverting the policy gradient is novel.

Weaknesses: 1. Perhaps the largest weakness is that the problem setting is new, so it's unclear how useful it is. I would increase my score if the paper were revised to include some more realistic experiments showing the utility of the problem setting (preferably, on real-world data). 2. A smaller concern is that the empirical evaluation on the Mujoco tasks don't actually show that the proposed method is learning the right reward function. Rather, they show that optimizing the inferred reward function yields high reward on the original reward function. I would increase my score if the paper were revised to show either (1) that the inferred reward function *weights* converged to the true values, or (2) that the method works well for different reward functions defined for the same dynamics.

Correctness: The theoretical results about estimating the gradient (Thm 4.1) seems correct, though I haven't read the proof in the appendix. The experimental protocol seems correct. Two small comments: * The claim that NN parameters are identifiable (L169) seems very likely to be violated, right? I think this merits a bit more discussion. * Another baseline would be to perform TREX [Brown 19] on the demos, assuming that each demo is better than the previous one.

Clarity: Yes, the paper clearly explains the problem setting and proposed method.

Relation to Prior Work: Yes, the discussion of related work is good.

Reproducibility: Yes

Additional Feedback: Questions: * L29: Is the claim that LOGEL will be more robust to out-of-distribution states validated empirically? If not, please remove this claim. * L138: To what distribution does the high-probability bound apply? If the observer gets to see the uncorrupted trajectories from the learner, there shouldn't be any noise, right? * L144 "intrinsic bias" -- Can you add a bit more explanation of this? * Eq 8 Can you explain where the \alpha_r > \epsilon constraint comes from? * Fig 3 "four learners": Where are the 4 learners coming from? * L254: "does not affect the correctness of the recovered weights." If it didn't affect the correctness of the weights, wouldn't the weight difference (Fig 1, far right) go to zero? * L280: Did the experiments only use policy iterates 10 - 20? * L286: To clarify, did the reacher and hopper experiments use the standard reward functions for all methods? If not, why? Minor comments * The motivation about multi-agent cooperation is quite nice. * "Reinforcement Learning" is not capitalized (though the acronym "RL" is capitalized). * L23 semicolon should be a comma * L36 "However" missing comma * L40 "that" -> "which" * S2: please be consistent about upper/lower-case for states $s$ * L73: If \phi is a vector, the range should be [-M_r, M_r]^q * L83: The input of f has dimension d+q, not d*q, right? * L127 "Obviously..." grammar error in this sentence. * L129 "them" ambiguous pronoun reference. * Alg 1: Where is this referred to in the main text? Also, sentences (lines 1-3) should end with periods. * L221 "In works on the theory of minds" -> "In work on theory of mind" * Fig 1 caption: "G(PO)MDP algorithm" -> "G(PO)MDP" * Fig 2, colorbar: please put the colors in sorted order (-5, -3, -1, 0, 7). * L234: "(1, 1)" -> "top left" [the coordinate system in ambiguous) * L245 (5,10,...) Add spaces between list items


Review 2

Summary and Contributions: Post-rebuttal: Thank you for your detailed responses. I still think that more baselines should be compared in the main body of the paper. There appears to be room in the plots for additional baselines. The two I would recommend would be behavioral cloning using the latest X trajectories from the learner and TREX or another preference-based learning algorithm. Also, better justification for the learning from a learner setting would strengthen this paper. On the whole, I think this paper is a solid contribution and explores an interesting problem space. --- This paper proposes a method for learning the reward function of a learning agent. In particular, the paper assumes that the learner is a gradient-based RL agent. By watching the agent improve at a task over time the proposed algorithm is able to estimate the true reward function of the learner. This allows the observing agent to optimize a good policy even if the learner is not yet optimal. This is an interesting new area of work that has potential applications in multi-agent systems or learning from watching a human improve at a task over time.

Strengths: This work presents a nice unified methodology and theory for learning from gradient learners. The theoretical results give insights into the sample complexity of the algorithms and the empirical results show that the method works well for simple gridworld tasks as well as mujoco tasks.

Weaknesses: The linearity assumption seems to be one of the biggest weaknesses. Prior work on learning from a learner such as Jacq et al. "Learning from a Learner" and Brown et al. "Extrapolating Beyond Suboptimal Demonstrations via Inverse Reinforcement Learning from Observations" do not make this assumption. It would be nice to show at least one experiment where the reward function non-linear to see how the linear assumption effects results. Another weakness is the requirement to run behavioral cloning to recover the learner's policy parameters. I would like to see more justification for why one can expect to have enough observation data to perform good behavioral cloning. How much training data is needed for the experiments. The paper needs more baselines. In particular, it would be useful to compare against behavioral cloning using the best data from the learner. Also, the T-REX algorithm by Brown et al. is mentioned in the appendix but I think the paper would benefit from adding both behavioral cloning and T-REX baselines to the main paper for the mujoco and gridworld tasks.

Correctness: The methodology and theoretical claims appear correct.

Clarity: The paper is well organized and well structured.

Relation to Prior Work: The paper states that the work by Jacq et al. is the only other work to study learning from a learner. However, this is not true. As mentioned in the appendix, Brown et al. "Extrapolating Beyond Suboptimal Demonstrations via Inverse Reinforcement Learning from Observations" also explicitly study the learning from a learning setting. This should be added to the related work section. Also another uncited, but very related work is "Inverse Reinforcement Learning from a Learning Agent" by Kubala, Konidaris, and Greenwald from RLDM 2019. (see paper #288 in http://rldm.org/papers/extendedabstracts.pdf)

Reproducibility: Yes

Additional Feedback: Figure 2 needs darker colors or maybe hatching to make the different colors obvious. In Figures 1 and 2 it would be good to also show the best performance of the learning agent to compare. Presumably LOGEL can learn a near optimal policy when the learner still has poor performance but this needs to be shown empirically. LOGEL assumes a linear reward function but I do not think that competing methods such as T-REX and LfL require this. It would be useful and interesting to compare the performance of T-REX and LfL when they are also given the assumption of a linear reward function and have access to the same linear reward features as LOGEL and just need to estimate the weights of the linear reward.


Review 3

Summary and Contributions: The authors propose a method for inverse reinforcement learning when the data generated comes from a gradient-based learner rather than an expert. The goal of this work is to learn the parameters of a linear reward function used by the learner. A connection between successor representation and the optimal parameters for this IRL problem is shown and used to propose a sample bound for the approximate case, when the gradient of the successor representation with respect to the learner's policy parameter is estimated from trajectories. A algorithm is proposed, LOGEL, which first tries to find the policy parameters used by the learner by optimizing the likelihood of the observed trajectories, then jointly optimizes the learning rates and the reward function parameters given the estimated policy parameters. The LOGEL algorithm is compared to an existing "Learning from a Learner" (LfL) method in a gridworld domain and two MuJuCo domains, Reacher and Hopper.

Strengths: The approach is sound and relevant to the IRL focused members of the NeurIPS community.

Weaknesses: I have concerns about the applicability of this approach. The assumption that the agent updating parameters according to the gradient of the return seems quite strong. There are many RL methods that don't necessarily do that, such as value-based methods. Additionally, many RL agents are online learners and don't fit into this framework of generating several trajectories using fixed policies. Not much can be done about latter but some experiments showing how performance degrades as this first assumption is violated would help establish to what degree this is a problem, if at all. The authors should compare with at least one similar IRL method (e.g., specialized for linear reward function) which is designed for the "learning from experts" setting. This paper is built around the idea that learning from learners provides a clear advantage in the IRL setting but this isn't obviously the case. Estimating the gradients of the successor representation required for this approach is not an easy problem. To be clear, I'm not trying to argue that the idea of learning from learners doesn't provide a benefit, but it also doesn't seem impossible to me that a "learning from experts" method would be competitive despite not leveraging the "learning from learner" setting, especially when considering the additional things LOGEL needs to compute/estimate.

Correctness: If I am understanding the MuJuCo experimental setup, the reward function for LfL doesn't have access to the very informative features that LOGEL is given. This doesn't seem like a fair comparison and could easily explain the difference in performance observed in the Hopper domain. It seems curious that LOGEL optimizing the learned reward function does better than the learner using the true rewards around 500k steps. With out an explanation, this makes me worry about the validity of those results. The theoretical results don't make any outlandish claims so I am inclined to believe that the results hold. It's hard to properly evaluate the results as there seems to be some missing assumptions and definitions in theorem 4.1. What is delta? How is Psi bounded? What is lambda_min, the smallest eigenvalue? The proofs are broken into numerous lemmas in the appendix and are mixed in with results that could be omitted. For this reason, I didn't not carefully verify the proof.

Clarity: The paper is mostly easy to follow but the paper would benefit from some proofreading to correct grammatical mistakes and instances of missing words, e.g., line 127.

Relation to Prior Work: A comparison to a "learning from experts" IRL method should be included.

Reproducibility: Yes

Additional Feedback: - It would help improve clarity if the quantities, such as m, d and q, were present in the theorem statement so that the reader doesn't need to go back and find them to understand the results. - The method used to compute the confidence interval should be mentioned. Post rebuttal ============== I greatly appreciate the additional results which address several of my concerns. I have increased my score to reflect this. I am still concerned about the significance of the MuJoCo results given the setup as I understand it even after the author's response. If the authors believe that my concerns on the subject are caused by confusion on my part, I would recommend they provide more details and justification for the MuJuCo experimental setup in the main paper.


Review 4

Summary and Contributions: Post-rebuttal: I appreciate the detailed author response! I feel that most of my concerns were addressed, and thus am increasing my score to a 6. I strongly encourage for these clarifications to be added to the paper, to make it more clear. In particular, in the paper it would help to expand the motivation for why learning from suboptimal trajectories is useful, and clarify that LOGEL does not require knowing the learner's policy structure. I'm still confused why for Hopper, the policy trained with the *ground-truth* reward function (i.e., the learner) does worse than the policy trained with the reward function that LOGEL finds (Figure 4, red versus blue line). Are these two reward functions substantially different, or is this difference just due to noise? ------------------- This paper proposes a new approach, called LOGEL, for the "learning from a learner" (LfL) task, in which the goal is for an observer to perform inverse reinforcement learning by watching a learner learn. LOGEL assumes that the learner is gradient-based. It takes as input a sequence of trajectory datasets generated by learner policies from consecutive learner steps, and uses behavioral cloning to estimate the learner's policy parameters at each learning step. Then these estimated policy parameters are used for jointly estimating the learning rates and reward parameters. The key difference between LOGEL and prior work on LfL is that LOGEL does not assume that the learner is monotonically improving with every learning update. The main contributions of this work are: the approach itself, a theoretical analysis of the sample complexity of this approach, and an empirical comparison of this approach against prior work, in three domains.

Strengths: One strength is that this work has strong theoretical grounding, in terms of how many examples from the learner are required in order to, with high probability, achieve a bound on the difference between the observer's estimate of the reward function parameters versus the true reward function. Another strength is that the empirical evaluation includes an evaluation of LOGEL for four types of learners, including non-gradient-based learners. Other strengths are that the approach is novel, the code for the submission is provided, and the Appendix contains details for reproducibility.

Weaknesses: I have several concerns about the proposed approach. First, the empirical results give mixed messages. In one out of three tasks (i.e., reacher), the LfL baseline significantly outperforms LOGEL (Figure 4, left). Whereas for another task (i.e., hopper), the policy trained with the reward function recovered by LOGEL outperforms the policy trained on the true reward function. How is that possible? And what kind of reward function does the LfL baseline recover for the hopper task, that leads to no learning at all? I would appreciate a deeper discussion regarding these unexpected results, and about what kinds of domains/tasks one would expect LOGEL to outperform LfL in. Second, when estimating the learner's policy parameters (Section 5.1), does this assume knowledge of the structure of the learner (e.g., the architecture of a neural network)? If so, this requirement is a limitation of the approach and should be clearly stated. If this is not a strict requirement, then the experiments should also evaluate the case where there is a mismatch between the structure of the learner's policy and the behavioral cloning policy. The third concern is regarding motivation and real-world applicability of LfL. I agree that inverse reinforcement learning is valid, because learning a reward function improves generalization to completely new situations, compared to directly learning a policy via behavioral cloning. However, I'm not convinced that it is more beneficial to learn this reward function by only observing how the learner improves early on in training, and ignoring the higher-reward trajectories that the learner obtains near the end of training. The Introduction explains what LfL is, but does not give the motivation behind why this is an impactful area to study further. Finally, given that the main advantage of LOGEL over prior work is that LOGEL doesn't assume the learner is monotonically improving, how likely is it that a gradient-based learner (with appropriately-chosen learning rates) will actually experience a substantial drop in performance during the early stages of training? I would like to see a citation for this, or some concrete examples of this given -- it's fine if those examples are in a toy domain.

Correctness: Yes, the claims, method, and empirical methodology seem correct.

Clarity: Overall the paper is clearly written. However, the Learning from a Learner (LfL) setting should be more clearly described. It's not clear that the learner's trajectories are taken from the beginning stages of learning, rather than near the end of learning. This is implied by the empirical evaluation, but never stated explicitly. It would help to include a figure that is similar to Figure 1 in Jacq et al., to communicate this clearly.

Relation to Prior Work: Yes. Although I would like a more clear explanation of how/why LOGEL performs better than preference-based and ranking-based IRL. The latter two are natural approaches for learning from trajectories taken from different points in learning.

Reproducibility: Yes

Additional Feedback: I appreciate the comparison to T-REX in the Appendix. I think There are some grammatical errors and typos in the paper, for example: - line 33: "Jacq et Al." --> "Jacq et al." - line 93: "relies in" --> "relies on" - line 93: "Recently was also proved that" is incorrect - line 127: "Obviously more data are available to estimate the gradient more accurate" --> "Obviously the more data are available to estimate the gradient, the more accurate"and useful

[Author Response · NeurIPS 2020]

We thank the reviewers for insightful comments and suggestions. We hope the rebuttal will clarify all the doubts.

**Motivation and real-world application** In the LfL [1,2] setting, we do not ignore the near-optimal trajectories,
but we can use all the training data to recover the reward function. In some cases, such as in multi-agent IRL,
the agent is interested in learning the rewards of other agents before the other agents become experts, to adopt
strategic behavior. Also in autonomous driving, when there is a new circuit, instead of waiting for the driver to
become an expert, we can learn his intentions in the initial learning phase. In the appendix, we reported an experi-
ment that uses the trajectories collected from an autonomous driving simulator (used in real-world problems [3]) to
recover the reward function. We think this experiment may show a more **realistic application** of the LfL problem.

**Gridworld** In the gridworld experiment, we show LOGEL's performance when the *gradient-*
*based learner assumption* is *violated*. We experiment with three no-gradient learners (Value
Iteration, Q-Learning, and Soft Policy Iteration) and one gradient learner (GPOMDP). The
results show that LOGEL works well even when the hypothesis is violated. In Figure 2) (10
runs) we show the performance of LOGEL using only the **online samples** of Q-Learning.

**MuJoCo** In these experiments, the learner uses the original reward function, but we construct
the reward features for LOGEL from the state and action features. In the Hopper environment,
we add the alive feature. The LfL algorithm uses, as in the original paper, a reward function
which is very informative as they model the reward with a policy that has the same structure
of the learner policy. In Figure 3) (6 runs) we show the behavior of LOGEL in the Hopper
environment without the alive feature, using only the l2 distance between the current state
and the next state and the norm, the squared norm, and the cubed norm of the action. The
good behavior of the learned policies with the recovered reward weights is because the optimal
reward function to learn a task can be different from the real one [5].

**R1** We answered questions 1 and 2 above. In the MuJoCo experiments, we use a different
reward space than the learner one, so we cannot show convergence to the original reward
weights. As suggested, we report another experiment with different reward features, Fig.3.
Other questions: L138 the noise is due to the estimation of the gradient from trajectories; the
intrinsic bias refers to the bias introduced by the estimated gradient that cannot be reduced
with more learning steps; $\alpha_r > \epsilon$ because the learning rate is by assumption greater than 0;
in gridworld experiment we used four learners: SVI, SPI, Q-learning and G(PO)MDP, we
collected their trajectories and we estimate the reward function with LOGEL; L254 at the end of
(paper) Fig.1 the weight difference is zero; L286 the Reacher and the Hopper experiments use
the original reward functions for the learner, but different reward features for LOGEL created
from the state and action information. In the appendix we compared LOGEL with TREX.

**R2** We can think that in the MuJoCo environment the linearity assumption is violated as we use
a different reward space for the recovered reward function. Behavioral cloning, as in LfL, is a
requirement for running our algorithm. However, we think that by using the behavioral cloning
of the previous iteration as a starting point, we can learn a good approximation of the current
policy after a few learning steps. In the MuJoCo experiment, and the Gridworld experiment with the GPOMDP learner,
we only use a subset of the demos that the learner uses to update her policy and we can assume that in a real-world
scenario we have access to such demos. Thank you for pointing out another related work, we will add the citation.

**R3** In the gridworld experiment, we use a value-based and a Q-Learning learner that violate the gradient-based
assumption. In Figure 2) we show the behavior of LOGEL using only the online updates of a QLearning learner. In
Figure 1) (10 runs), we compare the performance of LOGEL with GIRL [4], which is an IRL batch model-free algorithm.
The experiment shows that the algorithm cannot recover the correct reward weights from suboptimal trajectories. We
answer to the Hopper question in the MuJoCo section above. $\Psi$ is bounded in l2-norm, $\lambda_{\min}$ is the smallest eigenvalue,
the confidence interval is computed with t-distribution.

**R4** In the Hopper domain, as in the original paper [1], the LfL baseline does not recover a correct reward function; this
behavior is due to the starting behavior of the simulated robot which at first often falls to the ground; the LfL algorithm
tends to consider these absorbing states as good ones. The domain tasks in which LOGEL outperforms LfL are tasks
in which monotonous improvement is violated, in which there are absorbing states and where the learner is not a soft
policy improvement algorithm (as the gridworld experiment shows). In the behavioral cloning, we *do not assume the*
*knowledge of the policy* model of the learner: for example, in the gridworld environment, the Q-Learning and the SVI
learner are $\epsilon$-greedy actors, but we model their policies as Boltzman policies; also, in the appendix, in the autonomous
driving experiment, the learner has a rule-based policy, while we use a linear layer to approximate the policy. In Figure
1) we show how a batch-model free IRL algorithm cannot learn using only learning trajectories. We think that the
merits of LOGEL go beyond removing the monotonically improving assumption: we provide theoretical insights into
the correctness of the proposed method and we empirically show that the proposed method works well also with learner
using value-based methods (as value-iteration) and soft policy improvement algorithms, outperforming in most of the
experiments the LfL baseline.

[1] Jacq, A., Geist, M., Paiva, A., Pietquin, O. (2019). Learning from a Learner. ICML
[2] Kubala, V., Konidaris, G., Greenwald, A. (2019). Inverse Reinforcement Learning from a Learning Agent, RLDM
[3] Likmeta, A., Metelli, A. M., Tirinzoni, A., Giol, R., Restelli, M., Romano D., (2020) Combining reinforcement learning with rule-based controllers for transparent and general decision-making
in autonomous driving, Robotics and Autonomous Systems
[4] Pirotta, M. and Restelli, M. (2016). Inverse Reinforcement Learning through policy gradient minimization. AAAI
[5] Ng et al. "Policy invariance under reward transformations: Theory and application to reward shaping." ICML 1999


[Meta-Review · NeurIPS 2020]

Drawing upon Inverse RL, the submission proposes learning from an expert, which is using a learning process to optimize its reward. In the initial reviews, three of four reviewers were positive on the submission, and after seeing the author feedback, one of the reviewers was persuaded to raise the overall score, so that the current scores are now (7, 7, 6, 5). With these scores, it will be likely (but not guaranteed) to be accepted to NeurIPS. Regardless, it is important to, and we trust that you will, address all of the issues that were raised by the reviewers in the next version of the manuscript.